# Learning Normalized Energy Models for Linear Inverse Problems

**Nicolas Zilberstein** [1]   **Santiago Segarra** [1]   **Eero P. Simoncelli** [2 3]   **Florentin Guth** [2 3]

## Abstract

Generative diffusion models can provide powerful prior probability models for inverse problems in imaging, but existing implementations suffer from two key limitations: $(i)$ the prior density is represented implicitly, and $(ii)$ they rely on likelihood approximations that introduce sampling biases. We address these challenges by introducing a new energy-based model trained for denoising with a covariance-based regularization term that enforces consistency across different measurement conditions. The trained model can compute normalized posterior densities for diverse linear inverse problems, without additional retraining or fine tuning. In addition to preserving the sampling capabilities of diffusion models, this enables previously unavailable capabilities: energy-guided adaptive sampling that adjusts schedules on-the-fly, unbiased Metropolis-Hastings correction steps, and blind estimation of the degradation operator via Bayes rule. We validate the method on multiple datasets (ImageNet, CelebA, AFHQ) and tasks (inpainting, deblurring), demonstrating competitive or superior performance to established baselines. Code is available at https://github.com/nzilberstein/Anisotropic-energy-Model.

## 1. Introduction

Generative diffusion models (Sohl-Dickstein et al., 2015; Ho et al., 2020; Song & Ermon, 2019; Song et al., 2021b) have achieved remarkable success in image generative modeling. Beyond sampling, the primary use of such models is to provide prior probabilities for inverse problems (Daras et al., 2024), such as deblurring or inpainting. For this purpose, current solutions fall into two categories: *Bayesian*

methods (Kadkhodaie & Simoncelli, 2021; Kawar et al., 2022; Chung et al., 2023) combine the prior embedded in a pre-trained unconditional diffusion model, with measurement likelihoods via Bayes rule. While this separation offers flexibility, the likelihood term is intractable, and thus computing solutions relies on approximations that can lead to biased sampling (Chung et al., 2023; Bruna & Han, 2024). *Regression* methods (Saharia et al., 2022; Liu et al., 2023; Delbracio & Milanfar, 2023; Negrel et al., 2025; Elata et al., 2025) learn conditional diffusion models that generate clean signals directly from measurements. Such methods avoid likelihood approximations, but they sacrifice flexibility: they are trained to sample from the posterior density for a specific form of measurement, and must be retrained for each measurement model. In addition, both approaches are score-based, rendering posterior density evaluation computationally expensive.

We address these challenges by developing an energy-based model (EBM) that learns explicit and normalized prior and posterior densities while providing efficient access to both posterior means and posterior samples. A single trained model can be used to recover images corrupted by a diverse range of degradations in a consistent manner. This unified formulation, which leverages the connection between linear inverse problems and anisotropic denoising, also unlocks novel capabilities that we detail below. A number of previous publications have combined unnormalized EBMs with unconditional diffusion models (Du et al., 2023; Thornton et al., 2025), leveraging explicit energy values for compositional generation. While recent proposals use regularization terms that enable proper normalization (Guth et al., 2025; Yu et al., 2025; Plainer et al., 2025), these methods have so far been limited to isotropic noise and cannot properly handle the correlated noise that commonly arises when solving linear inverse problems.

Here, we introduce Anisotropic Covariance Score Matching (A-CSM), an extension of the "dual score-matching" framework of (Guth et al., 2025) that can handle anisotropic (colored) noise. We develop a novel regularizer that enforces consistency by constraining the gradient of the energy with respect to the noise covariance. Our primary contributions are:

- a novel training objective derived from the Fokker-

[1]Rice University, Houston, TX, USA [2]Flatiron Institute, New York, NY, USA [3]New York University, New York, NY, USA. Correspondence to: Nicolas Zilberstein <nzilberstein@rice.edu>.

*Proceedings of the $43^{rd}$ International Conference on Machine Learning*, Seoul, South Korea. PMLR 306, 2026. Copyright 2026 by the author(s).

Planck equation that enforces consistency of energies across noise covariance matrices;

- a complementary architecture that supports anisotropic noise scheduling via a novel noise embedding;

- experiments demonstrating that the resulting learned normalized densities enable new capabilities: energy-guided adaptive sampling, unbiased MCMC correction steps, and blind inverse problem solutions;

- experiments validating the model across multiple datasets (ImageNet, CelebA, AFHQ) and tasks (inpainting, deblurring), demonstrating that our single unified model can achieve performance competitive with or superior to established diffusion-based solvers.

## 2. Background and related work

### 2.1. Diffusion models

Diffusion models (Sohl-Dickstein et al., 2015; Ho et al., 2020; Song et al., 2021b) are derived from two continuous-time Markov processes: $(i)$ a forward process that gradually corrupts clean data with noise, and $(ii)$ a reverse process that generates samples by iterative denoising. Throughout this work, we assume a variance-exploding (VE) formulation of the forward process (Song et al., 2021b):

$$\mathbf{x}_t = \mathbf{x} + \sigma_t \mathbf{v}, \quad \mathbf{v} \sim \mathcal{N}(0, \mathbf{I}). \qquad (1)$$

The corresponding discrete reverse process is

$$\mathbf{x}_{t-1} = \mathbf{x}_t + \delta\sigma_t^2 \nabla \log p(\mathbf{x}_t) + \delta\sigma_t \mathbf{v}_t, \qquad (2)$$

where $\delta\sigma_t^2 = \sigma_t^2 - \sigma_{t-1}^2$ and the unknown score function $\nabla_{\mathbf{x}_t} \log p(\mathbf{x}_t)$ is approximated by a neural network $s_{\boldsymbol{\theta}}(\mathbf{x}_t, t)$ trained with denoising score matching (DSM) (Vincent, 2011; Raphan & Simoncelli, 2011). The DSM loss provides an upper-bound on $\mathrm{KL}(p(\mathbf{x}) \| p_{\boldsymbol{\theta}}(\mathbf{x}))$, and is therefore equivalent to maximum-likelihood training (Song et al., 2021a) for a particular weighting of the loss. Notably, the model provides an *implicit* representation of $p(\mathbf{x})$, represented using a family of score functions learned for all noise levels (Kadkhodaie & Simoncelli, 2021).

**Beyond isotropic noise.** While the majority of diffusion models assume an isotropic forward process, several recent studies have explored more general processes, including anisotropic processes with vector-valued parameterizations (Daras et al., 2023; Hoogeboom & Salimans, 2023; Bansal et al., 2023; Gerdes et al., 2024; Chen et al., 2024; Asthana et al., 2025) and learnable (Bartosh et al., 2024; Sahoo et al., 2024) and higher-order processes (Dockhorn et al., 2021; Singhal et al., 2023). Here, we make use of

anisotropic forward diffusion process with covariance schedule $(\boldsymbol{\Sigma}_t)_{t=0}^T$:

$$\mathbf{x}_t = \mathbf{x} + \boldsymbol{\Sigma}_t^{\frac{1}{2}} \mathbf{v}, \quad \mathbf{v} \sim \mathcal{N}(0, \mathbf{I}), \qquad (3)$$

with the corresponding reverse process

$$\mathbf{x}_{t-1} = \mathbf{x}_t + \boldsymbol{\delta\Sigma}_t \nabla \log p(\mathbf{x}_t) + \boldsymbol{\delta\Sigma}_t^{\frac{1}{2}} \mathbf{v}_t. \qquad (4)$$

### 2.2. Linear inverse problems

Linear inverse problems consist of estimating an unknown signal $\mathbf{x}$ from a noisy measurement $\mathbf{y}$, obtained via a degradation operator $\mathbf{H}$ (assumed linear):

$$\mathbf{y} = \mathbf{H}\mathbf{x} + \sigma\mathbf{v}, \quad \mathbf{v} \sim \mathcal{N}(0, \mathbf{I}). \qquad (5)$$

While most solutions aim for a point estimate (typically the posterior mean), one can also consider sampling from the posterior $p(\mathbf{x}|\mathbf{y}) \propto p(\mathbf{y}|\mathbf{x})p(\mathbf{x})$, where $p(\mathbf{y}|\mathbf{x})$ is the measurement model (5) and $p(\mathbf{x})$ is the prior implicitly embedded in a diffusion model. Two classes of solutions exist in the literature: *Bayesian approaches*, which approximate the prior $p(\mathbf{x})$ via a diffusion model and combine it with the measurement model at inference time; and *regression approaches*, which approximate the posterior $p(\mathbf{x}|\mathbf{y})$ directly through supervised regression on paired data $(\mathbf{x}, \mathbf{y})$.

**Bayesian approaches.** These methods generate a sample from the posterior by conditioning the reverse process (2) on the observation $\mathbf{y}$, through use of Bayes' rule:

$$\nabla_{\mathbf{x}_t} \log p(\mathbf{x}_t|\mathbf{y}) = \nabla_{\mathbf{x}_t} \log p(\mathbf{x}_t) + \nabla_{\mathbf{x}_t} \log p(\mathbf{y}|\mathbf{x}_t). \quad (6)$$

While the first term is typically computed using a pre-trained *isotropic* diffusion model, the second term (the so-called "guidance") is intractable, due to the need for high-dimensional integration: $p(\mathbf{y}|\mathbf{x}_t) = \int p(\mathbf{y}|\mathbf{x})p(\mathbf{x}|\mathbf{x}_t)\mathrm{d}\mathbf{x}$. Previous methods (Kadkhodaie & Simoncelli, 2021; Kawar et al., 2022; Chung et al., 2023) have resorted to approximations. But computational costs are high even for simple Gaussian approximations (Chung et al., 2023; Song et al., 2022), as the guidance term involves the *Jacobian* of the learned score. Beyond these guidance approaches, more recent studies have explored optimization-based approaches (Mardani et al., 2024; Zilberstein et al., 2025; Zhu et al., 2023; Feng et al., 2023), which frame posterior sampling as stochastic optimization; and sequential Monte Carlo-based (Wu et al., 2023) approaches, which leverage particle methods.

**Regression-based approaches.** A second category focuses on directly learning the conditional score (6). A diffusion model is trained to map noisy measurement $\mathbf{y}$ to the clean signal $\mathbf{x}$, effectively treating the inverse problem as

conditional generation by absorbing the degradation into the training data (Liu et al., 2023; Delbracio & Milanfar, 2023; Negrel et al., 2025; Hu et al., 2025). While these methods do not explicitly provide the degradation operator $\mathbf{H}$ to the score network, a few recent papers (Elata et al., 2025; Terris et al., 2026) have proposed degradation-aware parameterizations of the conditional denoiser $\mathbb{E}[\mathbf{x} \,|\, \mathbf{y}, \mathbf{H}]$. However, they rely on a conditioning mechanism that requires repeated mappings between the ambient and feature spaces at each level of the architecture, increasing the computational complexity.

# 3. Learning EBMs through anisotropic denoising

We aim to learn a *single* model $p_\theta(\mathbf{x}|\mathbf{y})$ that explicitly approximates the posterior *density* for measurements $\mathbf{y}$ arising from a variety of different degradation operators, in contrast to previous score-based approaches. Learning the posterior density gives access not only to the conditional score (6) through differentiation, but also unlocks several new applications that we describe in Section 4. We first describe in Section 3.1 the connection between linear inverse problems and anisotropic denoising, and how this can be leveraged to learn an energy model. We introduce our generalized dual score matching loss in Section 3.2 and our anisotropic energy architecture in Section 3.3. Finally, we show in Section 3.4 how our energy model can be used both as a denoiser and as a posterior sampler for linear inverse problems.

## 3.1. Energy-based linear inverse problem solvers

We leverage the equivalence between linear degradations and colored noise to show how an energy-based model conditioned on different noise covariances enables access to normalized posterior densities, posterior means, and posterior samples.

**From linear inverse problems to anisotropic denoising.**
Consider a linear inverse problem of the form $\mathbf{y} = \mathbf{H}\mathbf{x} + \sigma\mathbf{v}$. The posterior is unchanged by redefining $\mathbf{y}$ as $\mathbf{H}^{-1}\mathbf{y}$, which corresponds to observing

$$\mathbf{y} = \mathbf{x} + \mathbf{\Sigma}^{\frac{1}{2}}\mathbf{v}', \quad \mathbf{v}' \sim \mathcal{N}(0, \mathbf{I}), \qquad (7)$$

where $\mathbf{\Sigma} = \sigma^2 \mathbf{H}^{-1}(\mathbf{H}^{-1})^\top$. Linear inverse problems are thus equivalent to denoising problems for images contaminated with correlated additive noise, with the noise covariances $\mathbf{\Sigma}$ depending on the linear measurement $\mathbf{H}$. If $\mathbf{H}$ is not invertible, it can be stabilized by addition of a small multiple of the identity, which corresponds to adding large noise in the nullspace of $\mathbf{H}$. In the following, $\mathbf{y}$ refers to the observation model (7).

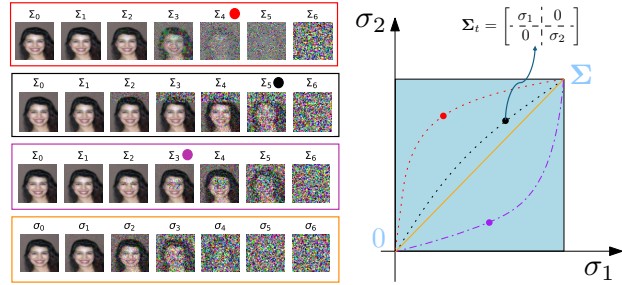

*Figure 1.* Illustration of possible paths in image space (left) and covariance space (right). Isotropic models are limited to covariance schedules that are reparameterizations of the diagonal (orange), while anisotropic models can explore paths in which different signal components are noise-corrupted at different rates (red, black, and purple).

**Anisotropic EBMs.** Using Bayes rule, the posterior distribution can be expressed as $p(\mathbf{x}|\mathbf{y}) = p(\mathbf{x})p(\mathbf{y}|\mathbf{x})/p(\mathbf{y})$. Normalization of this distribution requires having a model of $p(\mathbf{y})$, which depends on $\mathbf{\Sigma}$. To this end, we propose to learn an energy model $U_\theta(\mathbf{y}, \mathbf{\Sigma})$ that approximates $U(\mathbf{y}, \mathbf{\Sigma}) = -\log p(\mathbf{y}|\mathbf{\Sigma})$. Notice that we can recover $p(\mathbf{x})$ by setting $\mathbf{\Sigma} = 0$.

The posterior mean can be computed using an anisotropic generalization of Tweedie's identity: (Robbins, 1956; Miyasawa et al., 1961; Raphan & Simoncelli, 2011):

$$\mathbb{E}[\mathbf{x} \,|\, \mathbf{y}, \mathbf{\Sigma}] = \mathbf{y} - \mathbf{\Sigma}\nabla_\mathbf{y} U(\mathbf{y}, \mathbf{\Sigma}), \qquad (8)$$

where the score can be recovered from the energy model as $\nabla_\mathbf{y} U(\mathbf{y}, \mathbf{\Sigma}) \approx \nabla_\mathbf{y} U_\theta(\mathbf{y}, \mathbf{\Sigma})$. Therefore, by *learning an anisotropic denoiser*, we gain access to the posterior mean for different linear inverse problems.

Learning a score model over a family of covariances $\mathbf{\Sigma}$ enables posterior sampling. This is achieved by defining a noise schedule that bridges between measurements and the corresponding clean data. In particular, we need to specify a sequence of covariance matrices $(\mathbf{\Sigma}_t)_{t=0}^T$ that interpolate between the measurement covariance $\mathbf{\Sigma}$ at $t = T$ and the zero covariance at $t = 0$, as in (4). Several potential paths are illustrated in Figure 1. Note that unlike the isotropic setting, these schedules do not simply correspond to different discretizations of a correponsding continuous-time process.

## 3.2. Learning with anisotropic dual score matching

We learn a model of the energy $U(\mathbf{y}, \mathbf{\Sigma}) = -\log p(\mathbf{y}|\mathbf{\Sigma})$ by developing an anisotropic generalization of the (isotropic) dual score matching approach of Guth et al. (2025). In a nutshell, dual score matching consists of matching *both* derivatives of an energy model $U_\theta(\mathbf{y}, \mathbf{\Sigma})$ to the data, i.e., the gradients of the energy with respect to *(i)* the noisy input $\mathbf{y}$ (the "data score" used in diffusion models) and *(ii)* the covariance $\mathbf{\Sigma}$ (which we dub the "covariance score").

Each of these derivatives can be trained by minimizing a corresponding loss function. As we explain below, the dual score matching formulation enables learning of *normalized* densities.

**Data score.** The data score can be learned using the anisotropic generalization of Tweedie's formula (8), which leads to the anisotropic denoising score matching objective:

$$\ell_{A-DSM}(\theta) = \mathbb{E}\Big[\|\mathbf{\Sigma}^{\frac{1}{2}}\nabla_{\mathbf{y}}U_\theta(\mathbf{y},\mathbf{\Sigma}) - \mathbf{\Sigma}^{-\frac{1}{2}}(\mathbf{y}-\mathbf{x})\|^2\Big]. \tag{9}$$

The reweighting by $\mathbf{\Sigma}^{\frac{1}{2}}$ leads to a scale-invariant loss, a generalization of the maximum-likelihood weighting used in (Song et al., 2021a; Guth et al., 2025).

**Covariance score.** The covariance score can be shown to satisfy an analogue of the Tweedie formula (see Appendix A):

$$\nabla_{\mathbf{\Sigma}}U(\mathbf{y},\mathbf{\Sigma}) = \mathbb{E}\Big[\frac{1}{2}\mathbf{\Sigma}^{-1} - \frac{1}{2}\mathbf{\Sigma}^{-1}(\mathbf{y}-\mathbf{x})(\mathbf{y}-\mathbf{x})^{\top}\mathbf{\Sigma}^{-1}\Big].$$

This expression generalizes the time score identity of Guth et al. (2025); Yu et al. (2025); Plainer et al. (2025), and leads to a covariance score matching objective:

$$\ell_{A-CSM}(\theta) = \mathbb{E}_{\mathbf{x},\mathbf{y},\mathbf{\Sigma}}\Big[\Big\|\mathbf{\Sigma}^{\frac{1}{2}}\nabla_{\mathbf{\Sigma}}U_\theta(\mathbf{y},\mathbf{\Sigma})\mathbf{\Sigma}^{\frac{1}{2}} -$$
$$\frac{1}{2}\mathbf{I} + \frac{1}{2}\mathbf{\Sigma}^{-\frac{1}{2}}(\mathbf{y}-\mathbf{x})(\mathbf{y}-\mathbf{x})^{\top}\mathbf{\Sigma}^{-\frac{1}{2}}\Big\|_2^2\Big], \tag{10}$$

that has been weighted by $\mathbf{\Sigma}^{\frac{1}{2}}$ on both sides to be scale-invariant, and where $\|\cdot\|_2$ is the Frobenius norm. The A-CSM objective (10) acts as a regularizer that ties all the different marginals $p(\mathbf{y}|\mathbf{\Sigma})$ together in a meaningful way. It can be thought of as a proxy to enforce the continuity equation given by the Fokker-Planck equation (Pavliotis, 2014). This allows learning of a normalized density (as we show below), but also improves the denoising performance of the model, as illustrated in Appendix D.2.

**Normalization of the learned density model.** We define an overall objective as a weighted sum of the two objectives defined above: $\frac{1}{d}\ell_{A-DSM} + \frac{1}{d^2}\ell_{A-CSM}$ (see Appendix A). After training, the learned energy model $U_\theta(\mathbf{y},\mathbf{\Sigma})$ approximates the true energy up to a constant: $U_\theta(\mathbf{y},\mathbf{\Sigma}) \approx U(\mathbf{y},\mathbf{\Sigma}) + \text{cst}$. Importantly, the A-CSM objective ensures mass conservation through the space of covariances, and thus this constant is independent of $\mathbf{\Sigma}$. For large $\mathbf{\Sigma}$, $\mathbf{y}$ is approximately distributed as $\mathcal{N}(0,\mathbf{\Sigma})$, allowing to normalize the trained energy model as follows:

$$U_\theta(\mathbf{y},\mathbf{\Sigma}) \rightarrow U_\theta(\mathbf{y},\mathbf{\Sigma})$$
$$- \mathbb{E}_{\mathbf{y}}[U_\theta(\mathbf{y},\mathbf{\Sigma}) \mid \mathbf{\Sigma}] + \frac{1}{2}\log\det(2\pi e\mathbf{\Sigma}). \tag{11}$$

## 3.3. Architecture and covariance conditioning

We compute $U_\theta(\mathbf{y},\mathbf{\Sigma})$ with a neural network, with a novel architecture that is suitable for the problem. Since our proposed energy model is trained as an anisotropic denoiser, the first requirement is that the score $\nabla_{\mathbf{y}}U_\theta(\mathbf{y},\mathbf{\Sigma})$ derived from the energy preserves the inductive biases of score architectures used in previous literature to achieve high-quality denoising (and thus, score estimation). Following Romano et al. (2017); Guth et al. (2025); Thornton et al. (2025), we define

$$U_\theta(\mathbf{y},\mathbf{\Sigma}) = \frac{1}{2}\langle\mathbf{y},\mathbf{s}_{\boldsymbol{\theta}}(\mathbf{y},\mathbf{\Sigma})\rangle, \tag{12}$$

where $\mathbf{s}_\theta$ is an existing score network. Specifically, we implement $\mathbf{s}_\theta$ using the UNet architecture of Song et al. (2021b); Karras et al. (2022).

The second requirement is that the architecture should be able to operate under conditioning with a wide range of covariances $\mathbf{\Sigma}$ arising from different linear inverse problems. However, an arbitrary covariance has $d(d-1)/2$ degrees of freedom, which is problematic in terms of both memory and computational cost. To alleviate this, we limit ourselves to covariances that are diagonal in either the spatial or spatial frequency domain. This reduces the parameterization to $d$ coefficients, while still covering many standard inverse problems: inpainting (block-diagonal covariances in pixel space), deblurring (diagonal in frequency domain), and super-resolution (approximately diagonal in frequency domain).

Finally, we must define a mechanism by which the conditioning covariance matrix $\mathbf{\Sigma}$ is incorporated into the score network. Our design builds on the multiplicative conditioning mechanism used in isotropic score architectures via gain control (Karras et al., 2024). At each layer $\ell$, consisting of $c_\ell$ feature channels, an embedding vector $\mathbf{e}_\ell \in \mathbb{R}^{c_\ell}$ of the input noise variance $\sigma^2$ is computed and used to modulate (multiply) the channels. We represent spatial covariance matrices as spatially varying noise maps in $\mathbb{R}^d$, yielding *spatially varying embeddings* $\mathbf{e}_\ell \in \mathbb{R}^{c_\ell \times d_\ell}$, where $d_\ell$ denotes the spatial resolution at layer $\ell$. Spectral covariance matrices are incorporated through an analogous mechanism in the frequency domain, producing embeddings that modulate the corresponding feature channels (but not the spatial dimensions). At each layer, conditioning is applied through a gain modulation of the form $\mathbf{x}_\ell \leftarrow \text{SiLU}(\mathbf{x}_\ell \odot (1 + \mathbf{e}_\ell))$, where $\text{SiLU}(.)$ is the swish function (Elfwing et al., 2018). The resulting architecture is schematically illustrated in Fig. 2. Additional implementation details are provided in Appendix C.1. We emphasize that this new embedding module does not introduce significant additional computational overhead relative to those used for isotropic noise Karras et al. (2022).

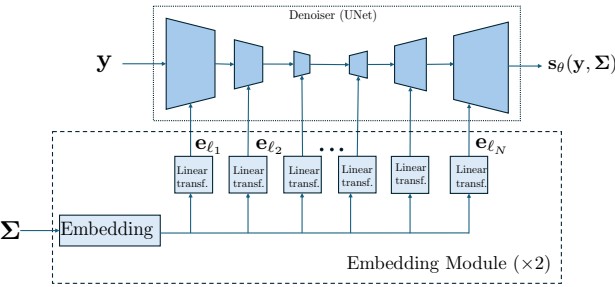

*Figure 2.* Proposed architecture based on UNet. We incorporate the covariance information through the embedding network, with two dedicated branches for the two covariance domains (spatial and spectral).

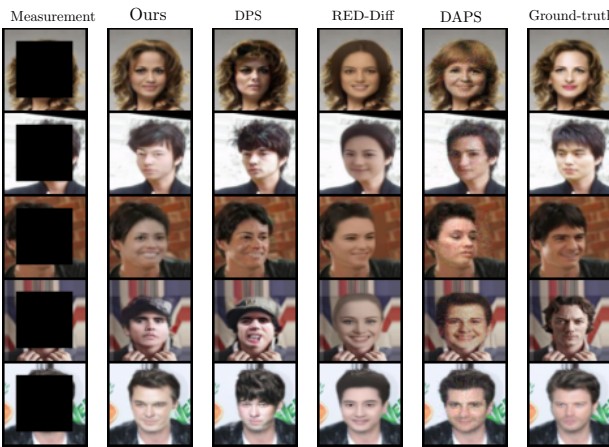

*Figure 3.* Inpainting results on CelebA. DPS and our method generate sharper images while RED-Diff's solutions are smoother, which is consistent with our analysis of their posterior density values in Section 4.1.

**Training and sampling.** We now describe our implementation and training pipeline. We assume a covariance distribution $p(\boldsymbol{\Sigma})$ for training in which the two classes (spatial and frequency covariances) are selected with probability $0.5$. In the spatial domain, we consider center-box and horizontal-box masks with sizes from 1 to $d^2 = 64^2$. In the spectral domain, we consider Gaussian deblurring and $\times 4$ super-resolution. Additional details regarding hyper-parameters and examples of the different covariances are provided in Appendix C, and the full training and sampling algorithms are provided in Appendix A.1.

### 3.4. Validation: comparison to baselines

**Experimental setting.** We compare our energy-based approach with several well-established methods (Daras et al., 2024). Among Bayesian baselines, we include the guidance methods DPS (Chung et al., 2023) and DAPS (Zhang et al., 2025a) and the variational inference-based RED-Diff (Mardani et al., 2024). We also include Palette (Saharia et al., 2022) as a conditional model trained to solve a particular inverse problem via regression. Details of each method are provided in Appendix C.4. We implement all methods, including our own, using the base architecture illustrated in Fig. 2. The differences lie in $p(\boldsymbol{\Sigma})$—Bayesian models have $\boldsymbol{\Sigma} = \sigma^2 \mathbf{I}$—and in the input provided to the model— conditional models stack the measurements with the input noisy image. Models were trained on CelebA ($64 \times 64$) (Liu et al., 2015), ImageNet64 ($64 \times 64$) (Russakovsky et al., 2015), AFHQ-Cat ($192 \times 192$) (Choi et al., 2020), and MNIST ($28 \times 28$) (LeCun et al., 2010).

We assess the reconstruction quality of the different samplers with mean squared error (in terms of PSNR $= -10 \log_{10} \text{MSE}$) and two perceptual metrics, LPIPS (Zhang et al., 2018) and DISTS (Ding et al., 2020). Each metric is averaged over 400 examples with ground truth images from the test set. In addition, we compute the FID (Heusel et al., 2017) between ensembles of test set images and posterior samples. All methods use up to $1,000$ neural function eval-

uations ($1,200$ for inpainting on CelebA). For each method, the number of steps is tuned via ablation and then fixed for all experiments; no ground-truth information is used at test time. This is especially relevant for RED-Diff, which performs better with fewer noise levels, consistent with Mardani et al. (2024).

**Experimental results.** We consider the two linear inverse problems that were used to jointly train the model. To test the spatial covariance configuration, we perform inpainting on a square of size $45 \times 45$ in the center of the image, with $\sigma = 10^{-4}$. For the spectral configuration, we perform deblurring with a Gaussian kernel of size $8 \times 8$ and a standard deviation of $0.8$, with $\sigma = 10^{-2}$. Results are shown in Table 1 for both CelebA and ImageNet, and in Fig. 3 we illustrate a few inpainting examples on CelebA. We observe that our proposed sampler is competitive with, and in most cases outperforms, the baseline samplers. We include additional comparisons on AFHQ-Cat ($192 \times 192$) (Choi et al., 2020) in Appendix D.1.3. Overall, these experiments demonstrate that energy models are fully competitive with or even improves on score models for solving linear inverse problems, and that a single trained model can be used to tackle a variety of inverse problems.

## 4. Applications of anisotropic energy models

Our energy model offers benefits beyond its use in solving linear inverse problems. We demonstrate in Section 4.1 that having access to the learned energy enables more refined comparisons between posterior sampling algorithms. In Sections 4.2.1 and 4.2.2, we develop samplers that leverage access to the energy to automatically define adaptive anisotropic schedules and use *unbiased* corrector steps via

*Table 1.* Quantitative results for inpainting and Gaussian deblurring across the CelebA and ImageNet64 datasets.

| Sampler | CelebA $64 \times 64$ | | | | | | | | ImageNet $64 \times 64$ | | | | | | | |
| | Inpainting | | | | Deblurring | | | | Inpainting | | | | Deblurring | | | |
| | PSNR↑ | LPIPS↓ | FID↓ | DISTS↓ | PSNR↑ | LPIPS↓ | FID↓ | DISTS↓ | PSNR↑ | LPIPS↓ | FID↓ | DISTS↓ | PSNR↑ | LPIPS↓ | FID↓ | DISTS↓ |
|---|---|---|---|---|---|---|---|---|---|---|---|---|---|---|---|---|
| DPS (Chung et al., 2023) | 16.10 | 0.110 | 36.76 | 0.19 | 32.98 | 0.004 | 43.01 | 0.08 | 23.01 | 0.051 | 55.61 | 0.114 | 29.10 | 0.007 | 59.09 | 0.10 |
| RED-Diff (Mardani et al., 2024) | **17.96** | 0.100 | 47.82 | 0.17 | **34.53** | 0.006 | 44.33 | 0.08 | **24.22** | 0.065 | 58.50 | 0.135 | **29.93** | 0.008 | 63.10 | 0.11 |
| DAPS (Zhang et al., 2025a) | 17.25 | 0.098 | 45.76 | 0.16 | 30.86 | 0.005 | 65.41 | 0.10 | 23.87 | **0.048** | 54.07 | 0.113 | 27.54 | 0.013 | 79.43 | 0.15 |
| Palette (Saharia et al., 2022) | 15.23 | 0.154 | 75.22 | 0.2 | 17.28 | 0.348 | 148.04 | 0.44 | – | – | – | – | – | – | – | – |
| **Ours** | 17.70 | **0.093** | **34.57** | **0.14** | 32.27 | **0.002** | **41.87** | **0.04** | 22.90 | 0.052 | **47.54** | **0.108** | 29.10 | **0.004** | **44.82** | **0.07** |

Metropolis-Hastings proposals. Finally, we show in Section 4.3 that our energy model can estimate unknown degradation operators and noise levels, enabling blind inverse problem solvers.

## 4.1. The probabilities of posterior samples

In this section, we compare the prior and posterior probabilities of samples generated by three solvers, which include a Bayesian model that relies on likelihood approximation (DPS), a variational inference procedure that approximates maximum-a-posterior optimization (RED-Diff), and a learned posterior sampler (ours). We fix one particular observation $\mathbf{y}$ and generate several potential solutions $\{\hat{\mathbf{x}}_i\}_{i=1}^N$ with each sampler. We then compute the probabilities of these samples under the prior $p(\mathbf{x})$ and the posterior $p(\mathbf{x}|\mathbf{y})$ (via Bayes rule) using our learned energy model. The resulting distributions of log probabilities are shown in Fig. 4. Examples of solutions at different prior probabilities are shown in the right panel: generally, smoother images tend to have higher probability, while more detailed, textured images have lower probability. In particular, we see that DPS generates samples that are lower-probability under the prior due to their likelihood approximations, whereas RED-Diff generates higher-probability samples due to its maximum-a-posterior behavior. Both methods produce samples of low posterior probability. Our energy-based sampler achieves a better balance: notably, the prior probabilities of solutions lie near the ground truth, and their posterior probability is only slightly lower than that of the ground truth. These results demonstrate that our anisotropic energy-based approach leads to accurate posterior samples that are consistent with both the prior and the measurements. A similar analysis across multiple observations $\mathbf{y}$ pairs is provided in Appendix D.3.

## 4.2. Energy-based schedules and corrector steps

### 4.2.1. Energy-guided generation

An important feature of anisotropic diffusion is the freedom it allows in designing sampling paths (Negrel et al., 2025; Gerdes et al., 2024), as illustrated in Fig. 1. For any sequence of covariance steps $\delta\boldsymbol{\Sigma}_t = \boldsymbol{\Sigma}_t - \boldsymbol{\Sigma}_{t-1}$, a sample can be generated with the iterations

$$
\begin{cases}
\mathbf{x}_{t-1} = \mathbf{x}_t - \delta\boldsymbol{\Sigma}_t \nabla_{\mathbf{y}} U_\theta(\mathbf{x}_t, \boldsymbol{\Sigma}_t) + (\delta\boldsymbol{\Sigma}_t)^{\frac{1}{2}} \mathbf{v}_t \\
\boldsymbol{\Sigma}_{t-1} = \boldsymbol{\Sigma}_t - \delta\boldsymbol{\Sigma}_t
\end{cases}
\tag{13}
$$

Designing samplers therefore reduces to selecting the *covariance step* $\delta\boldsymbol{\Sigma}_t$ at each iteration. After demonstrating this flexibility in the setting of any-order generation, we introduce an energy-guided sampler that selects $\delta\boldsymbol{\Sigma}_t$ automatically and adaptively by exploiting the variations of the energy with the noise covariance $\boldsymbol{\Sigma}$, leading to superior sampling quality.

**Any-order generation.** We consider a family of block-diagonal covariances in pixel space. Each covariance from the family has a fixed variance on each $b \times b$ image patch, where $b$ can take any value in $\{1, 2, 4, 7, 14, 28\}$ (see Fig. 9 in Appendix C.2 for an illustration). We trained a model on MNIST with this set of covariances, where the noise variance in each patch was independent and log uniformly distributed in $[10^{-9}, 10^3]$. This allows generating image patches in any order in an autoregressive manner. We illustrate three different orders with different values of $b$ in Fig. 5. Each path generates a different solution even with identical random seeds.

**Energy-guided adaptive sampler.** In theory, with a perfect score and an infinite number of sampling steps, all covariance schedules are equivalent. In practice, different schedules yield different approximations to the score and discretization errors. Which covariance paths lead to higher sampling quality, and how can we define them? We introduce a heuristically motivated *energy-guided* schedule that sets the covariance step $\delta\boldsymbol{\Sigma}_t$ adaptively by doing a descent step on the energy with respect to $\boldsymbol{\Sigma}_t$, using the covariance score $\nabla_{\boldsymbol{\Sigma}} U_\theta(\mathbf{x}_t, \boldsymbol{\Sigma}_t)$. Specifically, we set

$$
\delta\boldsymbol{\Sigma}_t \propto \boldsymbol{\Sigma}_t \nabla_{\boldsymbol{\Sigma}} U_\theta(\mathbf{x}_t, \boldsymbol{\Sigma}_t) \boldsymbol{\Sigma}_t.
\tag{14}
$$

The multiplication by $\boldsymbol{\Sigma}_t$ on both sides can be thought of as a preconditioning of the gradient (notice that $\delta\boldsymbol{\Sigma}_t$ has the same units as $\boldsymbol{\Sigma}_t$) and can be motivated as a steepest descent in the Bregman geometry generated by $-\log\det\boldsymbol{\Sigma}_t$ (see Appendix B). Equation (14) removes the need for choosing

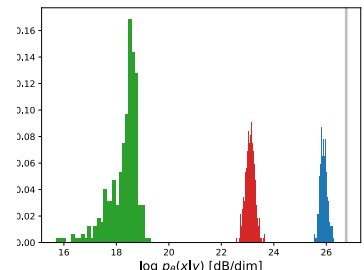
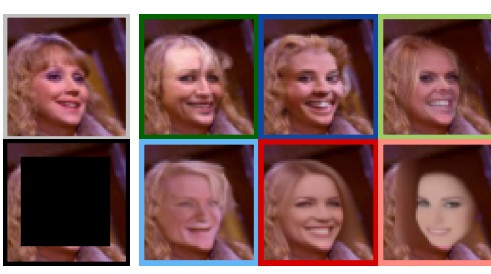

*Figure 4.* **Left and middle:** Histograms of $\log p_\theta(\hat{\mathbf{x}})$ and $\log p_\theta(\hat{\mathbf{x}}|\mathbf{y})$ for inpainting solutions $\hat{\mathbf{x}}$ generated by DPS, RED-Diff, and our energy model from a given measurement $\mathbf{y}$, along with the ground truth $\mathbf{x}$. Our energy model is well-calibrated with respect to both prior and posterior probabilities. **Right:** Examples of generated images $\hat{\mathbf{x}}$ sorted from lowest to highest prior probability (in reading order). The colored border indicates the sampler used to produce the image (green for DPS, red for RED-Diff, blue for ours) and its prior probability (darker shades for lower probability images), matching the arrows in the left panel. Note that higher-probability images are smoother and less detailed. The ground truth and measurements are shown on the left with blue/black borders, respectively.

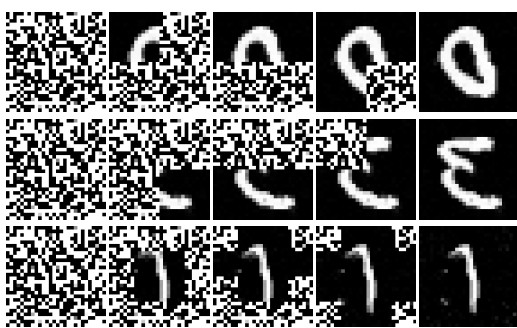

*Figure 5.* Three autoregressive generations for a model trained on MNIST, with identical initial and injected noise. The first two rows generate the four quadrants of the image respectively in reading and reversed reading order. The third row generates the image in 16 patches, ordered from the center to the outer border.

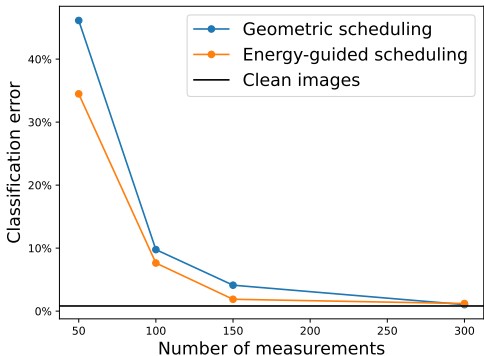

*Figure 6.* Classification error as a function of the number of measurements. The energy-guided scheduling (14, orange) consistently achieves a lower error rate than the fixed geometric scheduling $\delta\boldsymbol{\Sigma} \propto \boldsymbol{\Sigma}$ (blue) with few measurements, with both methods reaching the baseline classifier error at 300 measurements.

a discretization schedule during sampling and instead defines one automatically and adaptively to the current iterate $\mathbf{x}_t$, a property shared by the algorithm of Kadkhodaie & Simoncelli (2021) but in a different setting.

To illustrate the behavior of this sampler, we consider the task of reconstructing MNIST digits from a subset of $k$ randomly selected pixels (unobserved pixels are replaced by high-variance noise). Fig. 6 shows the classification error over $1,000$ samples as a function of the number of measurements $k$ (experimental details provided in Appendix C.1). Both samplers converge to clean-image baseline performance for large $k$, but the energy-guided sampler (14) achieves a better classification error than a geometric schedule $\delta\boldsymbol{\Sigma}_t \propto \boldsymbol{\Sigma}_t$ for smaller $k$. This illustrates the advantage of allowing the model to unmask different pixels at different rates in challenging inverse problems.

Preliminary experiments on a CelebA inpainting task indicate that the energy-guided sampler underperforms predefined schedules in this setting. We hypothesize that on more challenging tasks with more spatial variability, diagonal co-

variances in the spatial domain are not flexible enough to define meaningful paths. We believe that more general covariance spaces that are not restricted to a fixed basis should lead to gains similar to those we observe on MNIST, and leave exploration of those questions to future research.

### 4.2.2. Unbiased corrector steps with MALA

Typically, diffusion-based posterior samplers iterate between prediction and correction steps (Song et al., 2021b; Bradley & Nakkiran, 2025), where the correction is given by a version of the *Unadjusted Langevin Algorithm* (ULA):

$$\mathbf{x}'_t = \mathbf{x}_t - \frac{\eta}{2}[\nabla_\mathbf{y}U_\theta(\mathbf{x}_t, \boldsymbol{\Sigma}_t) - \nabla_{\mathbf{x}_t}\log p(\mathbf{x}_T|\mathbf{x}_t)] + \sqrt{\eta}\mathbf{v}'_t.$$
$$(15)$$

where $\mathbf{x}_T = \mathbf{y}$ and $p(\mathbf{x}_T|\mathbf{x}_t) \sim \mathcal{N}(\mathbf{x}_T; \mathbf{x}_t, \Sigma_T - \Sigma_t)$. Notice that we need an additional term in the correction step as we aim to sample from $p(\mathbf{x}_t|\boldsymbol{\Sigma}_t, \mathbf{y})$ rather than just $p(\mathbf{x}_t|\boldsymbol{\Sigma}_t)$. Since the energy model provides $U_\theta(\mathbf{x}_t, \boldsymbol{\Sigma}_t) \approx$

$-\log p(\mathbf{x}_t|\boldsymbol{\Sigma}_t)$, the likelihood term $\nabla_{\mathbf{x}_t}\log p(\mathbf{x}_T=\mathbf{y}|\mathbf{x}_t)$ acts as guidance to keep the iterates consistent with the observation $\mathbf{y}$ at covariance $\boldsymbol{\Sigma}_T$. It has been shown that using better MCMC samplers than ULA can boost the performance of diffusion-based samplers (Du et al., 2023). In this context, an advantage of our energy formulation is that it provides access to the density $p(\mathbf{x}_t|\boldsymbol{\Sigma}_t)$, in addition to the score, at every step. This enables the use of *unbiased* correctors such as the *Metropolis-Adjusted Langevin Algorithm* (MALA) (Roberts & Tweedie, 1996), which treats (15) as a proposal $q(\mathbf{x}_t'|\mathbf{x})$ that is accepted with probability $\min\left(1,\frac{p(\mathbf{x}_t'|\boldsymbol{\Sigma}_T,\mathbf{y})\,q(\mathbf{x}_t|\mathbf{x}_t')}{p(\mathbf{x}_t|\boldsymbol{\Sigma}_T,\mathbf{y})\,q(\mathbf{x}_t'|\mathbf{x}_t)}\right)$. We emphasize that the ability to compute this acceptance probability during *posterior* sampling is a distinctive feature of our anisotropic energy model, which explicitly provides access to the density $p(\mathbf{y}|\boldsymbol{\Sigma})$, setting it apart from prior isotropic energy-based diffusion approaches (Du et al., 2023; Thornton et al., 2025).

We report in Table 2 a comparison between ULA and MALA as a function of the number of corrector steps. We observe that increasing the number of MALA steps consistently improves reconstruction quality as measured by LPIPS with the ground truth, whereas ULA does not yield further gains. A visual comparison is provided in Fig. 29.

*Table 2.* LPIPS distance between ground truth $\mathbf{x}$ and reconstruction $\hat{\mathbf{x}}$ for different correction schemes on a CelebA inpainting task.

| Sampler | Number of corrector steps | | |
|---|---|---|---|
| | 1 | 5 | 8 |
| ULA corrector | 0.093 | 0.093 | 0.093 |
| MALA corrector | 0.093 | 0.091 | 0.089 |

### 4.3. Blind inverse problems

We show that our normalized energy model can be used to solve *blind inverse problems*, where the noise covariance $\boldsymbol{\Sigma}$ (or equivalently, the measurement operator $\mathbf{H}$) is unknown. Our proposed strategy is straightforward: we estimate $\boldsymbol{\Sigma}$ by maximizing the posterior probability $\log p(\boldsymbol{\Sigma}|\mathbf{y})=\log p(\mathbf{y}|\boldsymbol{\Sigma})+\log p(\boldsymbol{\Sigma})+\text{cst}$, given a prior on covariance matrices $p(\boldsymbol{\Sigma})$. Here, we choose a uniform prior over a set $\mathcal{S}$ of covariances, leading to

$$\hat{\boldsymbol{\Sigma}}=\underset{\boldsymbol{\Sigma}\in\mathcal{S}}{\arg\max}\,\log p_\theta(\mathbf{y}|\boldsymbol{\Sigma}). \qquad (16)$$

We illustrate this procedure on an inpainting task where the box size is unknown and is estimated through (16). More precisely, we observe $\mathbf{y}=\mathbf{x}+\boldsymbol{\Sigma}_s^{1/2}\mathbf{v}$, where $\boldsymbol{\Sigma}_s$ is a block-diagonal covariance with noise standard deviation $\sigma_1$ inside a central $s\times s$ box and $\sigma_2=10^{-4}$ outside of it. Fig. 7 shows that our energy model estimates accurately both the box size $s$ and the noise level $\sigma_1$. This success critically relies on the normalization of the energy values across covariances

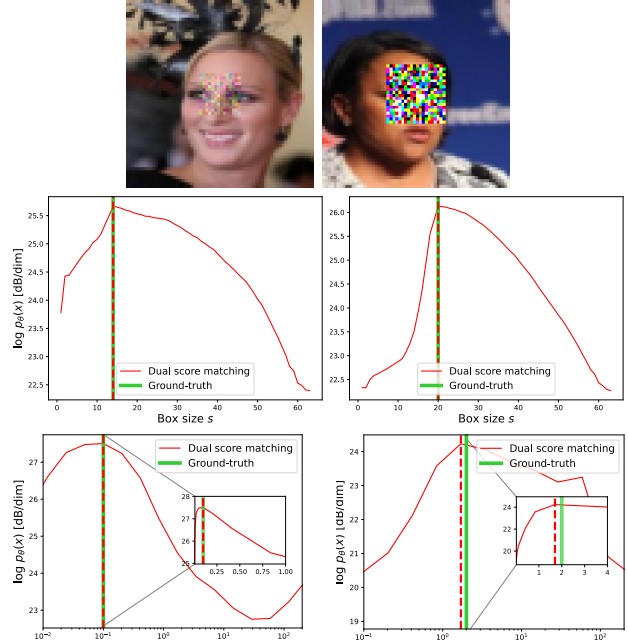

*Figure 7.* Blind reconstruction experiment. **Top:** two observations corrupted by noise of standard deviation $\sigma_1$ inside a central $s\times s$ square, with $(\sigma_1,s)\in\{(0.1,14),(2,20)\}$). **Middle and bottom:** Log probability $\log p_\theta(\mathbf{y}|\boldsymbol{\Sigma}_s)$ as a function of box size $s$ and $\sigma_1$, respectively. The noise covariance parameters estimated by the energy model trained with dual score matching (red dashed vertical lines) accurately track their ground truth values (solid green vertical lines).

$\boldsymbol{\Sigma}$: an energy model trained without the covariance score matching objective fails at this task.

## 5. Discussion

We have introduced a framework for learning normalized conditional densities that can be used to solve linear inverse problems. At the core of our method is Anisotropic Covariance Score Matching (A-CSM), a regularization that enforces consistency across covariance matrices via the Fokker-Planck equation, allowing accurate normalization of the learned density. We validate our approach on multiple datasets and inverse problems, demonstrating that covariance-conditioned energy-based models share the strong performance of current diffusion solvers while enabling new capabilities that score-based models do not have. In particular, explicit access to normalized densities enables blind estimation, energy-guided adaptive sampling, unbiased MCMC correction steps, and principled comparisons between posterior sampling algorithms.

It is useful to position our method within the literature on linear inverse problems, building on the distinction between Bayesian and regression-based methods introduced in Section 2.2. Bayes' rule, $p(\mathbf{x}|\mathbf{y})\propto p(\mathbf{x})\,p(\mathbf{y}|\mathbf{x})$, implies an

appealing separation between a prior model $p(\mathbf{x})$ and a likelihood term $p(\mathbf{y}|\mathbf{x})$: once a prior is learned, it can in principle be applied to any inverse problem with an explicit forward model without additional training. This approach however faces a fundamental challenge: computing posterior means $\mathbb{E}[\mathbf{x}\,|\,\mathbf{y}]$ or drawing posterior samples is intractable in high dimensions. For instance, for diffusion priors, posterior sampling requires the likelihood $p(\mathbf{y}|\mathbf{x}_t)$ at every noise level, which involves an intractable marginalization over the clean signal. The opposite approach—directly learning $\mathbb{E}[\mathbf{x}\,|\,\mathbf{y}]$, or more generally the conditional score $\nabla \log p(\mathbf{x}_t|\mathbf{y})$, via regression—can be both efficient and accurate, but it breaks the separation between an explicit prior shared across degradations and a problem-specific likelihood. As a result, modifying the likelihood term requires in principle additional training, and the relationships between the learned conditional expectation operators across different degradations are no longer explicit.

Within this context, our anisotropic energy model can be viewed as an intermediate approach that aims to reconcile these two viewpoints and combine their respective advantages of flexibility and tractability. Because it yields a learned normalized posterior density—and therefore the corresponding score function—it can be regarded as a generalization of the standard score-based methods discussed in Section 2.2. It also amounts to a change of perspective: rather than forming the posterior $p(\mathbf{x}|\mathbf{y})$ from a prior $p(\mathbf{x})$ and a likelihood $p(\mathbf{y}|\mathbf{x})$, we rely on the anisotropic diffusion (Fokker–Planck) equation $\nabla_{\boldsymbol{\Sigma}}\, p(\mathbf{y}|\boldsymbol{\Sigma}) = \frac{1}{2}\nabla_{\mathbf{y}}^2\, p(\mathbf{y}|\boldsymbol{\Sigma})$ to characterize the measurement density $p(\mathbf{y}|\boldsymbol{\Sigma})$, whose score gives efficient access to the posterior mean. The diffusion equation is enforced implicitly by the A-CSM objective, which guarantees consistency between a common prior and the posterior means associated with each $\boldsymbol{\Sigma}$. In this formulation, adding a novel degradation (i.e., covariance matrix $\boldsymbol{\Sigma}$) not seen during training requires additional learning to extend the range of $\log p(\mathbf{y}|\boldsymbol{\Sigma})$, but regularized by the A-CSM objective, leveraging what has already been learned.

Our method has several limitations. First, training energy-based models via (dual) score matching is more computationally intensive than training score-based models. We note that it could be accelerated using sliced score matching (Song et al., 2020), which enables replacing the extra backpropagation step with more memory-efficient (forward-mode) Jacobian vector products. Second, the spatial and spectral diagonal covariance parameterization covers a variety of standard inverse problems. Extending it to more general covariances would broaden the applications of our model and unlock the full potential of energy-guided sampling (at the cost of additional memory and computation).

We envision several applications of energy-based models that should be explored. First, they can provide estimates of posterior entropy and mutual information between measurements and ground truth signals, which could be used to optimize *measurement design*, offering a more direct method than Zhang et al. (2025b). Second, the dependence of the energy $U_\theta(\mathbf{y}, \boldsymbol{\Sigma})$ on the noise covariance $\boldsymbol{\Sigma}$ provides valuable information about the local geometry of the prior $p(\mathbf{x})$, allowing for example estimation of local curvature and "tangent subspaces" in the vicinity of individual images.

## Acknowledgments

The majority of this work was conducted during Nicolas Zilberstein's summer internship at Center for Computational Neuroscience, Flatiron Institute, a division of the Simons Foundation. The authors thank Jona Bruna, Florentin Coeurdoux, Pierre-Etienne Fiquet, Zahra Kadkhodaie, and Guy Ohayon for useful discussions. The authors thank the Scientific Computing Core, Flatiron Institute, for computing facilities and support. This research was partially sponsored by the Army Research Office under Grant Number W911NF-17-S-0002 and by the National Science Foundation under awards CCF-2340481 and EF-2126387. The views and conclusions contained in this document are those of the authors and should not be interpreted as representing the official policies, either expressed or implied, of the Army Research Office, the U.S. Army, or the U.S. Government. The U.S. Government is authorized to reproduce and distribute reprints for Government purposes, notwithstanding any copyright notation herein.

## Impact statement

This paper presents work whose goal is to advance the field of machine learning. There are many potential societal consequences of our work, none of which we feel must be specifically highlighted here.

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

# Appendices

## A. Dual-score matching for anisotropic noise

We expand on the dual-score matching formulation for anisotropic noise introduced in Section 3.2. Recall that we want to learn a covariance-dependent energy function, $U_\theta(\mathbf{y}, \boldsymbol{\Sigma}) = -\log p(\mathbf{y}|\boldsymbol{\Sigma})$, to approximate the negative log-likelihood (NLL) of the noisy image distribution, for a family of covariance matrices $\boldsymbol{\Sigma}$. We model this energy as follows

$$U_\theta(\mathbf{y}, \boldsymbol{\Sigma}) = -\log p(\mathbf{y}|\boldsymbol{\Sigma}) = -\log\left(\int p(\mathbf{x})p(\mathbf{y}|\mathbf{x}, \boldsymbol{\Sigma})\mathrm{d}\mathbf{x}\right) \tag{17}$$

To learn this model, we use dual score matching, which requires computing the gradients w.r.t. $\mathbf{y}$ and $\boldsymbol{\Sigma}$.

**Data score.**   We can use the Miyasawa-Tweedie identity (Miyasawa et al., 1961; Raphan & Simoncelli, 2011) given by

$$\nabla_\mathbf{y} U_\theta(\mathbf{y}, \boldsymbol{\Sigma}) = \mathbb{E}_\mathbf{x}\left[\boldsymbol{\Sigma}^{-1}(\mathbf{y} - \mathbf{x})|\mathbf{y}\right]. \tag{18}$$

We define the anisotropic denoising score matching loss as

$$\tilde{\ell}_{\mathrm{A-DSM}} = \mathbb{E}_{\mathbf{x},\mathbf{y}}\left[\|\nabla_\mathbf{y} U_\theta(\mathbf{y}, \boldsymbol{\Sigma}) - \boldsymbol{\Sigma}^{-1}(\mathbf{y} - \mathbf{x})\|_2^2\right] \tag{19}$$

To have a scale-invariant loss, we reweight by $\boldsymbol{\Sigma}^{\frac{1}{2}}$:

$$\ell_{\mathrm{A-DSM}} = \mathbb{E}\left[\|\boldsymbol{\Sigma}^{\frac{1}{2}}\nabla_\mathbf{y} U_\theta(\mathbf{y}, \boldsymbol{\Sigma}) - \boldsymbol{\Sigma}^{-\frac{1}{2}}(\mathbf{y} - \mathbf{x})\|^2\right]. \tag{20}$$

**Covariance score.** To obtain this second score, we differentiate the energy in (17) w.r.t. $\mathbf{\Sigma}$. Given that $p(\mathbf{y}|\mathbf{x}, \mathbf{\Sigma}) = e^{-\frac{1}{2}(\mathbf{y}-\mathbf{x})^\top \mathbf{\Sigma}^{-1}(\mathbf{y}-\mathbf{x}) - \frac{d}{2}\log(2\pi) - \frac{1}{2}\log\det(\mathbf{\Sigma})}$, we obtain

$$\nabla_{\mathbf{\Sigma}} U_\theta(\mathbf{y}, \mathbf{\Sigma}) = \mathbb{E}_{\mathbf{x}}[-\nabla_{\mathbf{\Sigma}} \log p(\mathbf{y}|\mathbf{x}, \mathbf{\Sigma})|\mathbf{y}, \mathbf{\Sigma}] \tag{21}$$

$$= \mathbb{E}_{\mathbf{x}}\left[-\nabla_{\mathbf{\Sigma}}\left(-\frac{1}{2}(\mathbf{y}-\mathbf{x})^\top \mathbf{\Sigma}^{-1}(\mathbf{y}-\mathbf{x}) - \frac{d}{2}\log(2\pi) - \frac{1}{2}\log\det(\mathbf{\Sigma})\right)\Big|\mathbf{y}, \mathbf{\Sigma}\right].$$

The two gradient terms can be simplified as follows:

$$\nabla_{\mathbf{\Sigma}}\left[-\frac{1}{2}(\mathbf{y}-\mathbf{x})^\top \mathbf{\Sigma}^{-1}(\mathbf{y}-\mathbf{x})\right] = \frac{1}{2}\mathbf{\Sigma}^{-1}(\mathbf{y}-\mathbf{x})(\mathbf{y}-\mathbf{x})^\top \mathbf{\Sigma}^{-1} \tag{22}$$

$$\nabla_{\mathbf{\Sigma}}\left[-\frac{1}{2}\log\det\mathbf{\Sigma}\right] = -\frac{1}{2}\mathbf{\Sigma}^{-1}. \tag{23}$$

Substituting (22) and (23) into (21) gives

$$\nabla_{\mathbf{\Sigma}} U_\theta(\mathbf{y}, \mathbf{\Sigma}) = \mathbb{E}_{\mathbf{x}}\left[\frac{1}{2}\mathbf{\Sigma}^{-1} - \frac{1}{2}\mathbf{\Sigma}^{-1}(\mathbf{y}-\mathbf{x})(\mathbf{y}-\mathbf{x})^\top \mathbf{\Sigma}^{-1}\Big|\mathbf{y}, \mathbf{\Sigma}\right]. \tag{24}$$

Finally, we define the anisotropic covariance score matching loss as

$$\tilde{\ell}_{\mathrm{A-CSM}} = \mathbb{E}_{\mathbf{x},\mathbf{y}}\left[\left\|\nabla_{\mathbf{\Sigma}} U_\theta(\mathbf{y}, \mathbf{\Sigma}) - \left(\frac{1}{2}\mathbf{\Sigma}^{-1} - \frac{1}{2}\mathbf{\Sigma}^{-1}(\mathbf{y}-\mathbf{x})(\mathbf{y}-\mathbf{x})^\top \mathbf{\Sigma}^{-1}\right)\right\|_F^2\right], \tag{25}$$

Similarly to the data score, we weight the loss by $\mathbf{\Sigma}^{\frac{1}{2}}$ on both sides to be scale-invariant:

$$\ell_{\mathrm{A-CSM}} = \mathbb{E}_{\mathbf{x},\mathbf{y},\mathbf{\Sigma}}\left[\left\|\mathbf{\Sigma}^{\frac{1}{2}}\nabla_{\mathbf{\Sigma}} U_\theta(\mathbf{y}, \mathbf{\Sigma})\mathbf{\Sigma}^{\frac{1}{2}} - \frac{1}{2}\mathbf{I} + \frac{1}{2}\mathbf{\Sigma}^{-\frac{1}{2}}(\mathbf{y}-\mathbf{x})(\mathbf{y}-\mathbf{x})^\top \mathbf{\Sigma}^{-\frac{1}{2}}\right\|_2^2\right]. \tag{26}$$

**Final loss.** Now that we have the dual-score matching loss, we define an overall objective by integrating over all possible covariance matrices $\mathbf{\Sigma}$. Directly extending the combined loss from the paper yields

$$\ell(\theta, \mathbf{\Sigma}) = \mathbb{E}_{\mathbf{\Sigma}}[k_1 \ell_{\mathrm{A-DSM}}(\theta, \mathbf{\Sigma}) + k_2 \ell_{\mathrm{A-CSM}}(\theta, \mathbf{\Sigma})] \tag{27}$$

where $k_1, k_2$ are constants that normalize each term with respect to the ambient dimension $d$. In particular, we consider $k_1 = 1/d$ and $k_2 = 1/d^2$:

$$\ell(\theta, \mathbf{\Sigma}) = \mathbb{E}_{\mathbf{\Sigma}}\left[\frac{1}{d}\ell_{\mathrm{A-DSM}}(\theta, \mathbf{\Sigma}) + \frac{1}{d^2}\ell_{\mathrm{A-CSM}}(\theta, \mathbf{\Sigma})\right] \tag{28}$$

While this loss is appealing, it is impractical due to the size of the covariance. As mentioned in Section 3.2, we consider a diagonal matrix instead.

**Diagonal $\mathbf{\Sigma}$ with independent parameters.** We consider diagonal matrices where $\mathbf{\Sigma} = \mathrm{diag}(\mathbf{\Phi})$ with $\mathbf{\Phi} = [\Phi_1, \ldots, \Phi_d]^\top$. Each parameter is independent with prior $p(\mathbf{\Phi}) = \prod_{i=1}^d p(\Phi_i)$, where $p(\Phi_i) \propto \Phi_i^{-1}$ for $\Phi_i \in [\Phi_{\min}, \Phi_{\max}]$.

To compute the gradient with respect to $\mathbf{\Phi}$, we utilize the chain rule. Note that $\frac{\partial \mathrm{vec}(\mathbf{\Sigma})}{\partial \Phi_i}$ is a vector of length $d^2$ with a 1 at the position corresponding to the $i$-th diagonal element $\Sigma_{ii}$, and 0 elsewhere. For a generic function $U_\theta(\mathbf{y}, \mathbf{\Sigma})$, the gradient component $i$ is:

$$[\nabla_{\mathbf{\Phi}} U_\theta(\mathbf{y}, \mathbf{\Sigma}(\mathbf{\Phi}))]_i = \mathrm{vec}(\nabla_{\mathbf{\Sigma}} U_\theta(\mathbf{y}, \mathbf{\Sigma}))^\top \cdot \frac{\partial \mathrm{vec}(\mathbf{\Sigma})}{\partial \Phi_i} \tag{29}$$

$$= [\nabla_{\mathbf{\Sigma}} U_\theta(\mathbf{y}, \mathbf{\Sigma})]_{ii} \tag{30}$$

Specifically, for the log-determinant term in (21), the gradient is:

$$\left[\nabla_{\mathbf{\Phi}}\left(-\frac{1}{2}\log\det\mathbf{\Sigma}\right)\right]_i = \mathrm{vec}\left(-\frac{1}{2}\mathbf{\Sigma}^{-1}\right)^\top \cdot \frac{\partial \mathrm{vec}(\mathbf{\Sigma})}{\partial \Phi_i} \tag{31}$$

$$= -\frac{1}{2}(\mathbf{\Sigma}^{-1})_{ii} = -\frac{1}{2\Phi_i} \tag{32}$$

Thus, in vector form, we obtain the simplified result:

$$\nabla_{\boldsymbol{\Phi}}\left[-\frac{1}{2}\log\det\boldsymbol{\Sigma}\right] = -\frac{1}{2}\boldsymbol{\Phi}^{-1} \tag{33}$$

For the quadratic term, we have the following (assuming $\mathbf{z} = (\mathbf{y} - \mathbf{x})$)

$$\begin{aligned}\nabla_{\boldsymbol{\Phi}}\left[-\frac{1}{2}(\mathbf{y}-\mathbf{x})^{\top}\boldsymbol{\Sigma}^{-1}(\mathbf{y}-\mathbf{x})\right] &= \text{vec}\left(\frac{1}{2}\boldsymbol{\Sigma}^{-1}\mathbf{z}\mathbf{z}^{\top}\boldsymbol{\Sigma}^{-1}\right) \cdot \frac{\partial\,\text{vec}(\boldsymbol{\Sigma})}{\partial\boldsymbol{\Phi}}, \\ &= \frac{1}{2}\,\text{vec}\big((\boldsymbol{\Phi}^{-1}\circ\mathbf{z})(\boldsymbol{\Phi}^{-1}\circ\mathbf{z})^{\top}\big) \cdot \frac{\partial\,\text{vec}(\boldsymbol{\Sigma})}{\partial\boldsymbol{\Phi}},\end{aligned} \tag{34}$$

where $\circ$ is the element-wise product. The $i$-th element is

$$\left[\nabla_{\boldsymbol{\Phi}}\left[-\frac{1}{2}(\mathbf{y}-\mathbf{x})^{\top}\boldsymbol{\Sigma}^{-1}(\mathbf{y}-\mathbf{x})\right]\right]_i = \left[\frac{1}{2}\,\text{vec}\big((\boldsymbol{\Phi}^{-1}\circ\mathbf{z})(\boldsymbol{\Phi}^{-1}\circ\mathbf{z})^{\top}\big)\right]_{ii}. \tag{35}$$

Combining both (35) and (33), we get

$$\nabla_{\boldsymbol{\Phi}}U_{\theta}(\mathbf{y}, \boldsymbol{\Sigma}(\boldsymbol{\Phi})) = \mathbb{E}_{\mathbf{x}}\left[\frac{1}{2}\boldsymbol{\Phi}^{-1} - \frac{1}{2}(\boldsymbol{\Phi}^{-1}\circ\mathbf{z})^2\Big|\mathbf{y}, \boldsymbol{\Sigma}(\boldsymbol{\Phi})\right]. \tag{36}$$

The training algorithm is described in Alg. 1. Notice we are using $\lambda = d^{-\frac{1}{2}}$ in (28), which corresponds to the loss in the main paper.

## A.1. Algorithms

---
**Algorithm 1** Training
---
**repeat**
    $\mathbf{x} \sim p(\mathbf{x})$ (This is the dataset given beforehand)
    $\boldsymbol{\Phi} \sim p(\boldsymbol{\Phi})$
    $\mathbf{v} \sim \mathcal{N}(0, \mathbf{I})$
    $\mathbf{y} = \mathbf{x} + \boldsymbol{\Sigma}(\boldsymbol{\Phi})^{1/2}\mathbf{v}$
    Compute $\ell_{\text{A}-\text{DSM}}(\theta, \boldsymbol{\Sigma}(\boldsymbol{\Phi})) = \frac{1}{N}\sum_{\mathbf{x},\mathbf{y}}\|\nabla_{\mathbf{y}}U_{\theta}(\mathbf{y}, \boldsymbol{\Sigma}(\boldsymbol{\Phi})) - \boldsymbol{\Phi}^{-1}\circ(\mathbf{y}-\mathbf{x})\|_{\frac{\boldsymbol{\Phi}}{d}}^2$
    Compute $\ell_{\text{A}-\text{CSM}}(\theta, \boldsymbol{\Sigma}(\boldsymbol{\Phi})) = \frac{1}{N}\sum_{\mathbf{x},\mathbf{y}}\big\|\nabla_{\boldsymbol{\Phi}}U_{\theta}(\mathbf{y}, \boldsymbol{\Sigma}(\boldsymbol{\Phi})) - \big(\frac{1}{2}\boldsymbol{\Phi}^{-1} - \frac{1}{2}(\boldsymbol{\Phi}^{-1}\circ(\mathbf{y}-\mathbf{x})^2\big)\big\|_{\frac{\boldsymbol{\Phi}^2}{d^2}}$
    Combine $\ell(\theta) = \ell_{\text{A}-\text{DSM}}(\theta, \boldsymbol{\Sigma}(\boldsymbol{\Phi})) + \ell_{\text{A}-\text{CSM}}(\theta, \boldsymbol{\Sigma}(\boldsymbol{\Phi}))$
    Perform one gradient descent step on $\ell(\theta)$ w.r.t. $\theta$
**until** Convergence

---

**Algorithm 2** Predictor–Corrector Sampling

**Require:** Noise schedule $\{\boldsymbol{\Sigma}_0, \ldots, \boldsymbol{\Sigma}_T\}$, number of corrector steps $n_{\text{steps}}$, energy model $U_\theta$, measurement $\mathbf{y}$, binary matrix with the structure of the degradation $\boldsymbol{\Sigma}_{id} = \text{clamp}(\mathbf{H}^{-1}, 0, 1)$, r, temperature $\eta$

1: Initialize $\mathbf{x}_T \leftarrow \mathbf{y}$
2: **for** $c = T - 1$ to $0$ **do**
3:     *// PREDICTOR STEP.*
4:     Compute score gradient $\mathbf{g}_c = -\nabla_{\mathbf{x}_c} U_\theta(\mathbf{x}_c, \boldsymbol{\Sigma}_c)$
5:     **if** Schedule = Adaptive **then**
6:         Compute covariance gradients $\mathbf{g}_{\Sigma_c} = -\nabla_{\boldsymbol{\Sigma}} U_\theta(\mathbf{x}_{c,0}, \boldsymbol{\Sigma}_c)$
7:         Covariance gradient descent + PSD constraint:

$$\boldsymbol{\Sigma}_{c+1} \leftarrow \max(\boldsymbol{\Sigma}_c - \eta_\Sigma \, \boldsymbol{\Sigma} \circ \mathbf{g}_\Sigma \circ \boldsymbol{\Sigma}, \varepsilon \mathbf{I})$$

8:         Define $\Delta\boldsymbol{\Sigma} = \boldsymbol{\Sigma}_c \circ \mathbf{g}_{\Sigma_c} \circ \boldsymbol{\Sigma}_c$
9:     **else if** Schedule = Fixed **then**
10:        $\Delta\boldsymbol{\Sigma} = \boldsymbol{\Sigma}_c - \boldsymbol{\Sigma}_{c+1}$
11:     **end if**
12:     Sample noise $\mathbf{z} \sim \mathcal{N}(0, \mathbf{I})$

$$\mathbf{x}_{c+1,0} \leftarrow \mathbf{x}_c + \Delta\boldsymbol{\Sigma} \circ \mathbf{g} + \Delta\boldsymbol{\Sigma}^{1/2} \circ \mathbf{z}$$

13:     *// CORRECTOR STEP.*
14:     **for** $s = 0$ to $n_{\text{steps}}$ **do**
15:         Compute score gradient $\mathbf{g}_{c+1,s} = -\nabla_{\mathbf{x}_{c+1,s}} U_\theta(\mathbf{x}_{c+1,s}, \boldsymbol{\Sigma}_{c+1})$
16:         Sample noise $\mathbf{z} \sim \mathcal{N}(0, \mathbf{I})$
17:         **if** Domain = pixel **then**
18:             Compute step size $\epsilon \leftarrow \left(\frac{r \cdot \|\boldsymbol{\Sigma}_{id} \circ \mathbf{z}\|}{\|\boldsymbol{\Sigma}_{id} \circ \mathbf{g}\|}\right)^2 \cdot 2$
19:             Construct element-wise step size $\boldsymbol{\Sigma}_\epsilon = \boldsymbol{\Sigma}_{id} \circ \epsilon$

$$\mathbf{x}_{c+1,s+1} \leftarrow \mathbf{x}_{c+1,s} + \boldsymbol{\Sigma}_\epsilon \circ \mathbf{g}_{c+1,s} + \sqrt{2\,T}\, \boldsymbol{\Sigma}_\epsilon^{1/2} \circ \mathbf{z}$$

20:         **else if** Domain = frequency **then**
21:             Compute step size $\epsilon \leftarrow \left(\frac{r \cdot \|\mathbf{z}\|}{\|\mathbf{g}\|}\right)^2 \cdot 2$

$$\mathbf{x}_{c+1,s+1} \leftarrow \mathbf{x}_{c+1,s} + \epsilon(\mathbf{g}_{c+1,s} + \nabla_{\mathbf{x}_t} \log p(\mathbf{x}_T = y | \mathbf{x}_{c+1,s})) + \sqrt{2\,T}\, \epsilon \mathbf{z}$$

22:         **end if**
23:         **if** MALA **then**
24:             Compute Metropolis–Hastings acceptance probability $\alpha = \min\left(1, \frac{\exp\left(-U_\theta(\mathbf{x}^\star, \boldsymbol{\Sigma}_{c+1})\right) q(\mathbf{x}_{c+1,s} | \mathbf{x}^\star)}{\exp\left(-U_\theta(\mathbf{x}_{c+1,s}, \boldsymbol{\Sigma}_{c+1})\right) q(\mathbf{x}^\star | \mathbf{x}_{c+1,s})}\right)$
25:             Sample $u \sim \mathcal{U}(0,1)$ and accept/reject
26:

$$\mathbf{x}_{c+1,s+1} \leftarrow \begin{cases} \mathbf{x}^\star, & u < \alpha \\ \mathbf{x}_{c+1,s}, & \text{otherwise} \end{cases}$$

27:         **end if**
28:     **end for**
29: **end for**
    Return $\mathbf{x}_{0,n_{steps}}$

## B. Energy-guided sampling as a Bregman mirror descent

While the gradient $\nabla_{\boldsymbol{\Sigma}} U_\theta(\mathbf{y}, \boldsymbol{\Sigma})$ corresponds to the steepest descent with respect to the Euclidean norm, it is more natural here to consider the steepest descent with respect to the Bregman divergence generated by the negative log determinant:

$$D(\boldsymbol{\Sigma} \,\|\, \boldsymbol{\Sigma}') = \mathrm{tr}(\boldsymbol{\Sigma}'^{-1}\boldsymbol{\Sigma}) - \log\det(\boldsymbol{\Sigma}'^{-1}\boldsymbol{\Sigma}) - d. \tag{37}$$

Note that this Bregman divergence corresponds to twice the KL divergence between two zero-mean Gaussian distributions with covariance $\boldsymbol{\Sigma}$ and $\boldsymbol{\Sigma}'$. For a stepsize parameter $\gamma > 0$, a Bregman mirror descent step computes

$$\underset{\boldsymbol{\Sigma}'}{\arg\min}\; U_\theta(\mathbf{y}, \boldsymbol{\Sigma}') + \frac{1}{\gamma} D(\boldsymbol{\Sigma}' \,\|\, \boldsymbol{\Sigma}) = (\boldsymbol{\Sigma}^{-1} + \gamma\nabla_{\boldsymbol{\Sigma}} U_\theta(\mathbf{y}, \boldsymbol{\Sigma}))^{-1} \tag{38}$$

$$= \boldsymbol{\Sigma} - \gamma\boldsymbol{\Sigma}\nabla_{\boldsymbol{\Sigma}} U_\theta(\mathbf{y}, \boldsymbol{\Sigma})\boldsymbol{\Sigma} + o(\gamma), \tag{39}$$

where the last step is a first-order Taylor approximation for small stepsizes $\gamma \to 0$. This leads to the update

$$\delta\boldsymbol{\Sigma} = \boldsymbol{\Sigma}\nabla_{\boldsymbol{\Sigma}} U(\mathbf{y}, \boldsymbol{\Sigma})\boldsymbol{\Sigma}. \tag{40}$$

## C. Implementation

### C.1. Architecture and training hyperparameters

Our architecture is a modification of the SongNet NCSNPP network (Song et al., 2021b) implemented in Karras et al. (2022). We use the same hyperparameters as the original implementation, which are reported in Table 3; the only modification is the noise embedding, which we describe below. We also describe in Table 4 the hyperparameters of the training. The MNIST classifier used in Figure 6 is a CNN with 3 layers and ReLU activation functions.

**Noise Embedding Modulation.** Given the diagonal noise of the covariance $\mathrm{diag}(\boldsymbol{\Sigma}_t)$, we first compute embeddings:

$$\mathbf{e}_{\mathrm{spatial}} \leftarrow f_{\mathrm{spatial}}(\mathrm{diag}(\boldsymbol{\Sigma}_t)) \in \mathbb{R}^{B \times c_{\mathrm{emb}} \times H \times W}, \tag{41}$$

$$\mathbf{e}_{\mathrm{frequency}} \leftarrow f_{\mathrm{frequency}}(\mathrm{diag}(\boldsymbol{\Sigma}_t)) \in \mathbb{R}^{B \times c_{\mathrm{emb}} \times 1 \times 1}, \tag{42}$$

where $f_{\mathrm{spatial}}(.)$ and $f_{\mathrm{frequency}}(\boldsymbol{\Sigma}_t)$ are two residual architectures (ResNet) (He et al., 2015), consisting of $3 \times 3$ convolutions paired with Group Normalization and SiLU activations, and a residual connection. Given that the main architecture (a UNet) performs down and up-sampling operations both in the channel and in the spatial domain, we need to accommodate the dimensions of the embedding network to each layer $\ell$. For the channel dimension, we use $1 \times 1$ convolutional layers as follows:

$$\mathbf{e}_{\ell,\mathrm{spatial}} \leftarrow \mathrm{Conv}_{1 \times 1}(\mathbf{e}_{\mathrm{spatial}}) \in \mathbb{R}^{B \times c_\ell \times H \times W} \tag{43}$$

$$\mathbf{e}_{\ell,\mathrm{frequency}} \leftarrow \mathrm{Conv}_{1 \times 1}(\mathbf{e}_{\mathrm{frequency}}) \in \mathbb{R}^{B \times c_\ell \times 1 \times 1} \tag{44}$$

For the spatial dimensions, we use the following operations

$$\mathbf{e}_{\ell,\mathrm{spatial}} \leftarrow \mathrm{Interpolate}(\mathbf{e}_{\ell,\mathrm{spatial}}) \in \mathbb{R}^{B \times c_\ell \times H_\ell \times W_\ell} \tag{45}$$

Given the embedding vector for each layer, we apply a multiplicative conditioning mechanism, given by

$$\mathbf{x}_\ell = \mathrm{SiLU}(\mathrm{GroupNorm}(\mathbf{x}_\ell) \odot (1 + \mathbf{e}_{\ell,\mathrm{spatial/frequency}})) \tag{46}$$

where $\mathrm{GroupNorm}(.)$ is the group normalization, with a number of channels in each group of $\min(32, c_\ell/4)$.

*Table 3.* SongUNet Architecture Summary

| Variable/Module | Configuration |
|---|---|
| *Network Parameters* | |
| Image Resolution | $64 \times 64$ |
| Input/Output Channels | 3 |
| Model Channels | 128 |
| Channel Multipliers | [1, 2, 2, 2] |
| Embedding Channels | 512 ($128 \times 4$) |
| Number of Blocks per Resolution | 4 |
| Attention Resolutions | [16] |
| Dropout | 0.10 |
| *Embedding Configuration* | |
| Embedding Type | Positional |
| Noise Channel Multiplier | 2 |
| Noise Embedding Channels | 256 ($128 \times 2$) |
| Anisotropic Noise | True |
| *Anisotropic Gamma Embedding Network* | |
| Fourier Dimension | 256 |
| Base Channels | 128 |
| Residual Blocks | 4 |
| Output Channels | 512 |
| Time Range | $[10^{-9}, 10^3]$ |
| *Encoder Architecture* | |
| Resolution Levels | 64, 32, 16, 8 |
| Channels per Level | 128, 256, 256, 256 |
| Encoder Type | Standard |
| *Decoder Architecture* | |
| Decoder Type | Standard |
| Skip Connections | Yes |
| Adaptive Scaling | False |
| Resampling Filter | [1, 3, 3, 1] |
| *UNet Block Configuration* | |
| Attention Heads | 1 (at $16 \times 16$) |
| Skip Scale | $\sqrt{0.5}$ |
| Normalization | GroupNorm ($\epsilon = 10^{-6}$) |

*Table 4.* Training hyperparameters for CelebA and ImageNet64

| Hyperparameter | Value |
|---|---|
| Number of GPUs | 6 |
| Dataset | CelebA/ImageNet64 |
| Training batch size | 128 |
| $\sigma^2_{min}$ | $1 \times 10^{-9}$ |
| $\sigma^2_{max}$ | $1 \times 10^3$ |
| Learning rate | $2 \times 10^{-4}$ |
| Total steps | 400,000 |
| LR decay | 100,000 steps |
| Warmup steps | 1,000 |
| Grad. clipping | Norm = 20 |

*Table 5.* Training hyperparameters for AFHQ

| Hyperparameter | Value |
|---|---|
| Number of GPUs | 8 |
| Dataset | AFHQ ($192 \times 192$) |
| Training batch size | 64 |
| $\sigma^2_{min}$ | $1 \times 10^{-9}$ |
| $\sigma^2_{max}$ | $1 \times 10^3$ |
| Learning rate | $2 \times 10^{-4}$ |
| Total steps | 100,000 |
| Warmup steps | 1,000 |
| Grad. clipping | Norm = 20 |

## C.2. Distribution covariances used for training

As described in Section 3.3, we used four types of covariances: Gaussian deblurring, shown in Fig. 8a, super-resolution (a combination of Gaussian kernel with a downsampling operation), and box and half-mask inpainting, illustrated in Fig. 8b. The family of autoregressive patch-based covariances used in the MNIST experiments in Section 4.2.1 are shown in Fig. 9.

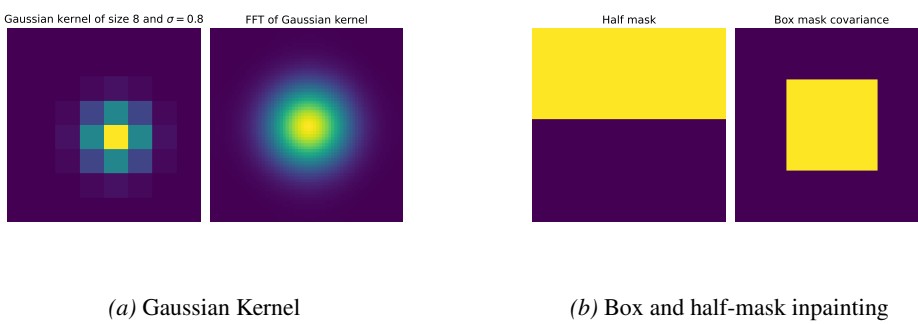

*(a)* Gaussian Kernel          *(b)* Box and half-mask inpainting

*Figure 8.* Examples of particular $\Sigma_t$ for different kernels and masks.

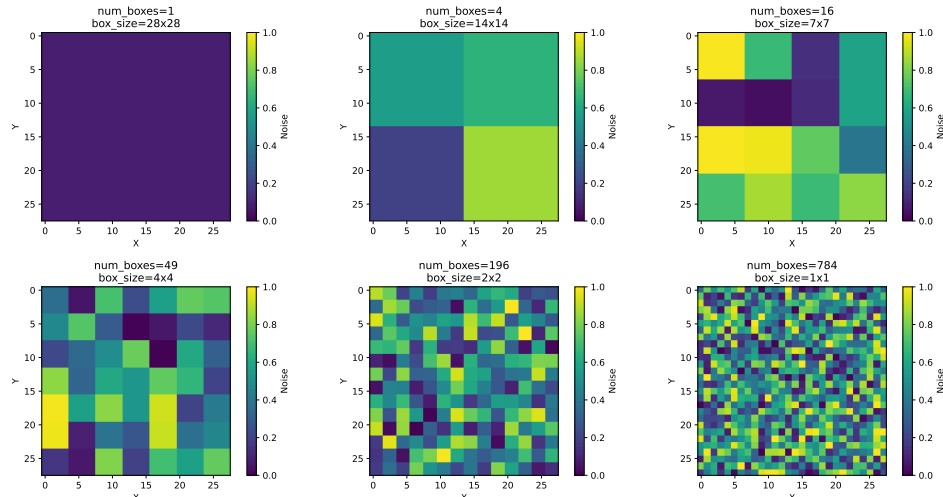

*Figure 9.* Examples of autoregressive patch-based covariances $\Sigma_t$ used for training. We first sample the number of boxes, and then the variance of each box.

## C.3. Comparison with previous energy-based models

Our method is more computationally intensive to train than standard score-based models. This overhead primarily comes from double backpropagation, which is inherent to energy-based formulations and is widely acknowledged in prior work (Thornton et al., 2025; Balcerak et al., 2025; Ai et al., 2026; Du et al., 2023). In this sense, the increased cost is not specific to our method, but rather a known trade-off when learning energy models. That said, we further contextualize below that our model has a computational cost comparable to prior work when accounting for model size and resolution.

1. We use the Song architecture from EDM ($\approx 70M$ parameters), which is substantially larger than the network used in the original DSM paper (Guth et al., 2025) ($\approx 13M$ parameters). The larger architecture naturally increases training time, while also yielding improved performance. For reference, EDM training at $64 \times 64$ takes approximately 4 days on 8 V100 GPUs ($\approx 4$ A100s) with a batch size of 256 (see `https://github.com/NVlabs/edm`). Scaling to $192 \times 192$ increases the pixel count by $9\times$, which significantly increases the computational cost, either in terms of GPU count or total training time.

2. This computational challenge is shared by energy-based models more broadly:

- Thornton (Thornton et al., 2025) used 8 A100 GPUs at $64 \times 64$ resolution (without the dual objective).
- Balcerak (Balcerak et al., 2025) trained at $64 \times 64$ on CelebA with 4 A100s and a batch size of 32; neither (Thornton et al., 2025) nor (Balcerak et al., 2025) report total training time.
- Boffi (Ai et al., 2026) required approximately 8 days for the flow matching teacher and 6 days for the shortcut model on ImageNet $64 \times 64$.
- Du (Du et al., 2023) trained at $128 \times 128$ using TPU clusters, making direct comparison difficult.

### C.4. Details of baselines

**RED-Diff.** It is a variational framework that frames the sampling problem as stochastic optimization by minimizing the KL divergence

$$q(\mathbf{x}_0|\mathbf{y}) = \underset{q(\mathbf{x}_0|\mathbf{y})}{\operatorname{argmin}} \operatorname{KL}(q(\mathbf{x}_0|\mathbf{y})||p(\mathbf{x}_0|\mathbf{y})). \tag{47}$$

When $q(\mathbf{x}_0|\mathbf{y}) \sim \mathcal{N}(\boldsymbol{\mu}, \sigma^2 \mathbf{I})$, the gradient update boils down to

$$\nabla_{\boldsymbol{\mu}} = \mathbb{E}_{t \sim \mathcal{U}[0,T]}\left[\nabla_{\boldsymbol{\mu}}\|\mathbf{y} - \mathbf{H}\boldsymbol{\mu}\|^2 + \mathbb{E}_{\epsilon \sim \mathcal{N}(0,I)}[\lambda_t(\boldsymbol{\epsilon}_{\boldsymbol{\theta}}(\mathbf{x}_t, \sigma_t) - \boldsymbol{\epsilon})]\right]$$

where $\boldsymbol{\epsilon}_{\boldsymbol{\theta}}(\mathbf{x}_t, \sigma_t) = -\sigma_t \mathbf{s}_{\boldsymbol{\theta}}(\mathbf{x}_t, \sigma_t)$ and $\mathbf{x}_t = \alpha_t \boldsymbol{\mu} + \sigma_t \boldsymbol{\epsilon}$; in the signal domain, we have $(\boldsymbol{\epsilon}_{\boldsymbol{\theta}}(\mathbf{x}_t, \sigma_t) - \boldsymbol{\epsilon}) \propto (\boldsymbol{\mu} - \mathbb{E}[\mathbf{x}_0|\mathbf{x}_t])$. In this formulation, the full trajectory acts as multi-scale regularizer, imposing coarse-to-fine details. We adapted the implementation from the original paper (Mardani et al., 2024) to our code. We follow the same weighting scheme for inpainting, and we use $\lambda = 0.25$ and $l_r = 0.1$, while for Gaussina deblurring we use $\lambda = 2$. As mentioned in Section 3.4, we train from scratch an isotropic version of the proposed network.

**DPS.** We adapted the implementation of the original paper (Chung et al., 2023) to our code. We tried using the same weight for the gradient of the likelihood, but for our implementation and a box size of 45, the sampler was diverging. Hence, we perform a grid search to get the best performance, with 0.05 the best one for inpainting, while 0.01 for deblurring. For all the experiments, we match the number of steps than our energy-guided sampler to have a fair comparison. We use the same base network as RED-Diff.

**DAPS.** We also implemented DAPS (Zhang et al., 2025a), which alternates between ODE-based denoising and Langevin dynamics for measurement consistency at each noise level. At each step, a short ODE flow (5 steps) estimates the clean image $\hat{x}_0$, followed by Langevin sampling that minimizes both the measurement residual and a prior term anchored to $\hat{x}_0$. We use 100 Langevin steps per noise level with step size $\eta = 2 \times 10^{-6}$, and 250 (for inpainting) and 200 (for deblurring) number of ODE runs. We use the same base network as RED-Diff.

**Palette.** We follow the formulation from the original paper (Saharia et al., 2022). Hence, we modified the Song architecture by stacking the noisy image $\mathbf{x}_t$ with the measurement $\mathbf{y}$, and using the same base architecture that we used for the guidance methods.

**Ours.** We describe the setup for our method in Table 6.

*Table 6.* Inpainting - Hyperparameters for the Predictor-Corrector Sampler with fixed corrector.

| Parameter | Value | Description |
|---|---|---|
| $\sigma_{\text{begin}}$ | 25 | Initial (maximum) noise standard deviation inside the mask. |
| $\sigma_{\text{end}}$ | $10^{-3}$ | Final (minimum) noise standard deviation inside the mask. |
| $T$ | 600 | Number of discretized noise levels. |
| $\sigma_{\text{dist}}$ | Geometric | Progression type of the noise schedule. |
| $N_{\text{steps}}$ | 1 | Number of corrector steps per noise level. |
| r | 0.15 | step-size for the Langevin update. |
| $\zeta$ (temp) | 0.9 | Temperature scaling for score-based sampling. |

**Computational cost for inference.** We expand on the running times for Gaussian deblurring across all methods, considering the running time for sampling a batch of 16 images at $64 \times 64$:

| Sampler | Time [sec] | Computational cost (NFE) |
|---------|-----------|--------------------------|
| Ours | 139 | $(N_p)$ NFEs $+ (N_p)$ grad w.r.t. input for energy |
| DPS | 167 | $N$ NFEs $+ N$ grad w.r.t. denoiser |
| RED-Diff | 22 | $N$ NFEs |

Importantly, the inference time of our method is comparable to DPS and standard diffusion samplers, so the computational overhead is concentrated entirely in training and does not affect sampling time. In contrast to these methods, our energy formualtion learns a full posterior density. We believe this makes the overhead modest relative to the added capabilities our approach unlocks, such as blind estimation and normalization constant computation, which are not possible with DPS or RED-Diff.

For sampling, the additional $\nabla_y U_\theta$ gradient is similar in cost to DPS's backpropagation and architectural overhead is minimal. Thus, wall-clock time is approximately equal at the same NFEs. For AFHQ $192 \times 192$: DPS 55 sec, ours 74 sec.

# D. Additional experiments

## D.1. Additional inverse problems

### D.1.1. In-distribution covariances

We include the results for **super-resolution** $\times 4$ and **horizontal half-mask inpainting** in Table 7; visual results can be found in Appendix D.6.

For super-resolution, we do not use corrector steps; in terms of numerical results, our method outperforms Bayesian approaches by a significant margin, similarly to the deblurring case. As illustrated in Fig. 26, the generated images present some artifacts. We attribute this partly to the known limitations of the frequency-domain conditioning: our frequency-domain degradations are defined with a Fourier transform with periodic boundary conditions, which can cause border effects. For practical use, a more careful implementation with more advanced boundary handling (e.g., a suitable discrete cosine transform) should be preferred.

Regarding half-mask inpainting, our method achieves better FID than baselines. However, in contrast to box inpainting, our method have a similar LPIPS than Bayesian baselines.

*Table 7.* Experiments on CelebA $64 \times 64$: Comparison on super-resolution $\times 4$ and half-mask inpainting.

| Sampler | Super-resolution $\times 4$ | | | Half-mask Inpainting | | |
|---------|:------:|:------:|:------:|:------:|:------:|:------:|
| | **PSNR↑** | **LPIPS↓** | **FID↓** | **PSNR↑** | **LPIPS↓** | **FID↓** |
| DPS (Chung et al., 2023) | 15.38 | 0.09 | 74.31 | 13.99 | 0.11 | 29.22 |
| RED-Diff (Mardani et al., 2024) | 15.22 | 0.08 | **64.75** | **15.65** | 0.10 | 34.84 |
| **Ours** | **22.41** | **0.01** | 93.72 | 14.00 | **0.10** | **28.15** |

### D.1.2. Out-of-distribution covariances

In this subsection, we illustrate how our method can handle out-of-distribution degradations, i.e., those inverse problems with an associated covariance $\Sigma$ not used for training. In particular, we consider **random inpainting**, where we mask out a 70% of pixels, **vertical half mask inpainting**, with a mask of size of $64 \times 30$, and **motion deblurring**; for the latter, we considered the kernel from https://github.com/LeviBorodenko/motionblur of size $15 \times 15$ and intensity 0.3, and $\sigma_v = 2 \times 10^{-3}$; and for scheduling, we use a Gaussian deblurring kernel of size $15 \times 15$ and intensity 0.8, which serves as an approximation. The results are shown in Table 8 and Table 9. Remarkably, we observe that our models achieves a similar performance than Bayesian methods.

*Table 8.* Experiments on CelebA $64 \times 64$: Comparison on random inpainting (0.7%) and vertical half-mask inpainting.

| Sampler | Random Inpainting (0.7%) | | | Vertical-half Mask | | |
|---|---|---|---|---|---|---|
| | PSNR↑ | LPIPS↓ | FID↓ | PSNR↑ | LPIPS↓ | FID↓ |
| DPS (Chung et al., 2023) | 29.58 | **0.01** | **21.83** | 14.21 | 0.085 | **31.54** |
| RED-Diff (Mardani et al., 2024) | **30.25** | **0.01** | 22.20 | **15.80** | **0.083** | 38.78 |
| **Ours** | 28.90 | **0.01** | 23.54 | 13.26 | 0.090 | 32.22 |

*Table 9.* Experiments on CelebA $64 \times 64$: Comparison on motion deblurring.

| Sampler | PSNR [dB]↑ | LPIPS↓ | FID↓ | DISTS↓ |
|---|---|---|---|---|
| DPS | 28.63 | 0.02 | 45.65 | 0.094 |
| RED-Diff | **31.00** | 0.02 | **44.33** | 0.085 |
| **Ours** | 29.42 | **0.007** | 50.6 | **0.083** |

### D.1.3. Large-scale experiments

In this subsection, we include experiments on AFHQ-Cat $192 \times 192$, demonstrating that our proposed model can be trained on large-scale images. To simplify training, we consider only inpainting with a center-box of sizes $s \times s$, with values from $s = 20$ to $s = 50$, and half-mask with sizes $s \times 64$, for the same values of $s$.

**Comparison with baselines.** We compare with DPS for a box inpainting task; the result is shown in Tables 11 and 13.

*Table 10.* Box $30 \times 30$

| Sampler | Inpainting (box of $30 \times 30$) | | | |
|---|---|---|---|---|
| | PSNR [dB] ↑ | LPIPS ↓ | FID ↓ | DISTS ↓ |
| Ours | 33.78 | **0.005** | **3.04** | **0.0095** |
| DPS | 31.91 | 0.007 | 3.59 | 0.0113 |
| RED-Diff | **37.58** | 0.008 | 4.81 | 0.0128 |

*Table 11.* Experiments on CelebA $64 \times 64$: Comparison on box inpainting $30 \times 30$.

*Table 12.* Box $50 \times 50$

| Sampler | Inpainting (box of $30 \times 30$) | | | |
|---|---|---|---|---|
| | PSNR [dB] ↑ | LPIPS ↓ | FID ↓ | DISTS ↓ |
| Ours | 26.45 | **0.021** | **9.43** | **0.029** |
| DPS | 25.21 | 0.023 | 10.71 | 0.031 |
| RED-Diff | **28.22** | 0.028 | 13.28 | 0.037 |

*Table 13.* Experiments on CelebA $64 \times 64$: Comparison on box inpainting $50 \times 50$.

**Blind experiment.** We also consider a blind experiment, following the setup from Section 4.3. In particular, we consider $\sigma_2 = 10^{-4}$, and $\sigma_1, s = \{1, 30\}$. The result is illustrated in Fig. 11; the value of the estimated $\sigma_1$ is 0.2.

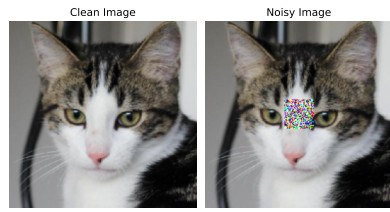

*Figure 10.* Noisy observation with a box covariance of size $30 \times 30$ and $\sigma_1 = 1$.

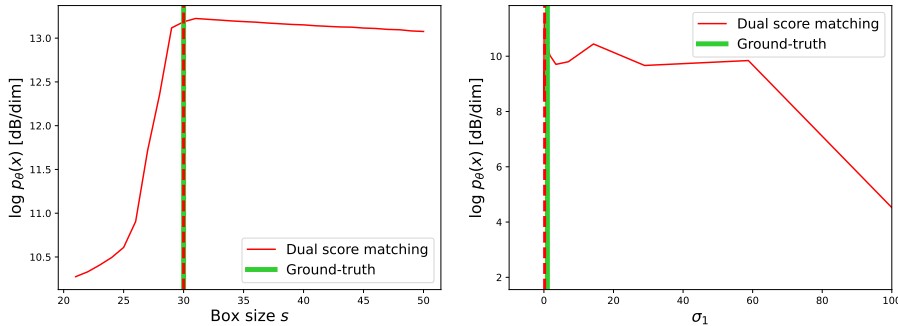

*Figure 11.* Blind reconstruction experiment. Log probability $\log p_\theta(\mathbf{y}|\boldsymbol{\Sigma}_s)$ as a function of box size $s$ and $\sigma_1$, respectively. The energy model trained with dual score matching correctly identifies the true box size (dashed red for estimated size and solid green vertical lines).

## D.2. Comparison between dual and single score matching

**Blind experiment.** Validating the correctness of the dual score matching is difficult since we do not have access to the the true density. However, we can use some experiments to assess this, like the blind experiment in Section 4.3. In particular, this experiment aims to find the covariance that best explains a given measurement, relying on a proper density across different covariances. While in Section 4.3 we ony compared the dual score matching with the ground truth, here we add the single objective to show that, in this additional case, the estimation fails; this is illustrated in Fig. 12.

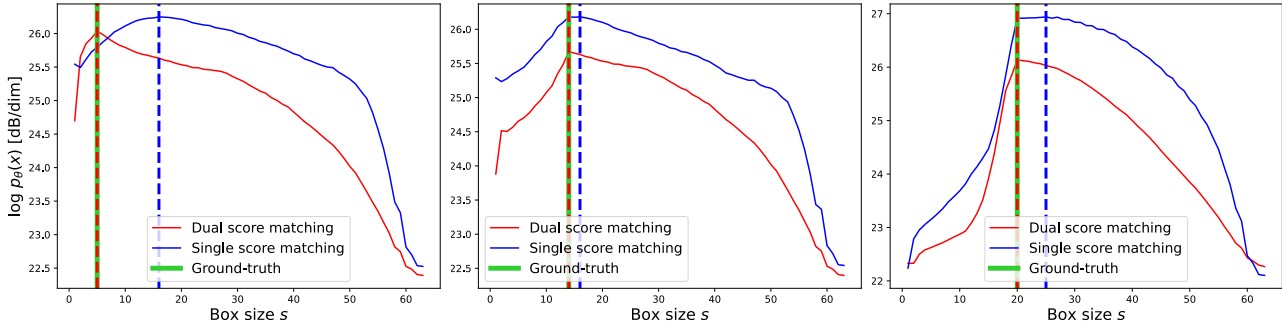

*Figure 12.* Log probability $\log p_\theta(\mathbf{y}|\boldsymbol{\Sigma}_s)$ as a function of box size $s$ for $s = \{5, 14, 20\}$ with $\sigma_1 = \{2, 0.1, 2\}$ respectively. The energy model trained with dual score matching correctly identifies the true box size (dashed red for estimated size and solid green vertical lines), which is not the case for the model trained with single score matching (dashed blue vertical line).

**One-shot conditional MMSE estimator.** One advantage of our proposed method is that we can compute $\mathbb{E}[\mathbf{x}|\mathbf{y}, \boldsymbol{\Sigma}]$ in one shot. We illustrate this for a denoising task with box of size $20 \times 20$ (the outside does not have noise), where we evaluate the one-shot denoiser for multiple noise level in the box. The ablation is shown in Fig. 13, where we compute the MSE as a function of $\sigma_1^2$, the noise inside the box. In this experiment, we also consider our proposed energy model trained with the single objective (only A-DSM). From this experiment, we observe that the energy-model trained with the dual-objective yields a better model in terms of denoising performance, showcasing that the regularization not only allows learning a normalized density model, but also helps for improving the denoising performance. To complement this experiment, we

illustrate a few visual examples in Fig. 14 using our energy model trained with the dual objective, and for a covariance with box of size $10 \times 10$, $\sigma_1 = 0.5$ (inside the box) and $\sigma_2 = 0.1$ (outside the box).

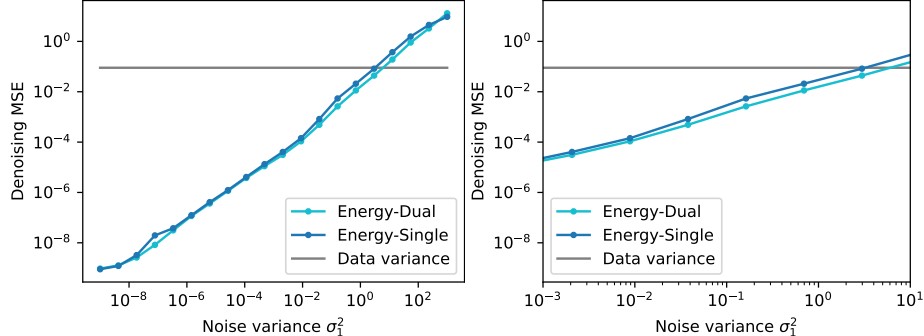

*Figure 13.* MMSE for one-shot denoising in box inpainting $(10 \times 10)$ as a function of the noise in the box.

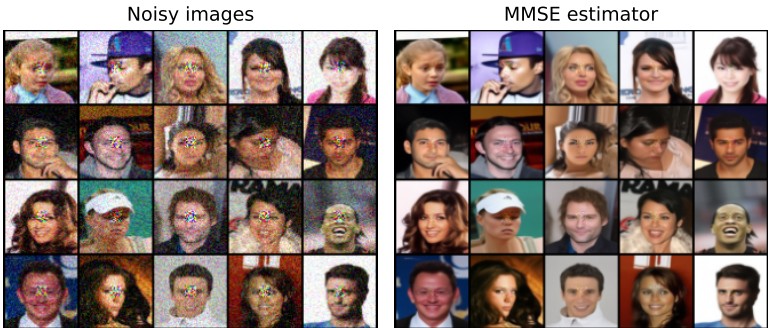

*Figure 14.* MMSE estimator computed by our model, which corresponds to $\hat{\mathbf{x}} = \mathbb{E}[\mathbf{x}|\mathbf{y}, \boldsymbol{\Sigma}]$.

### D.3. Analysis of the posterior distribution

**CelebA.** We include here additional results for the analysis of the posterior distribution described in Section 4.1. We start showing the distribution across multiple $(\mathbf{x}, \mathbf{y})$ in Fig. 15, with generated images sorted from high to low probability in Fig. 16. Notice that the behavior is similar to the single case, where our energy model is the one closer the posterior (left figure).

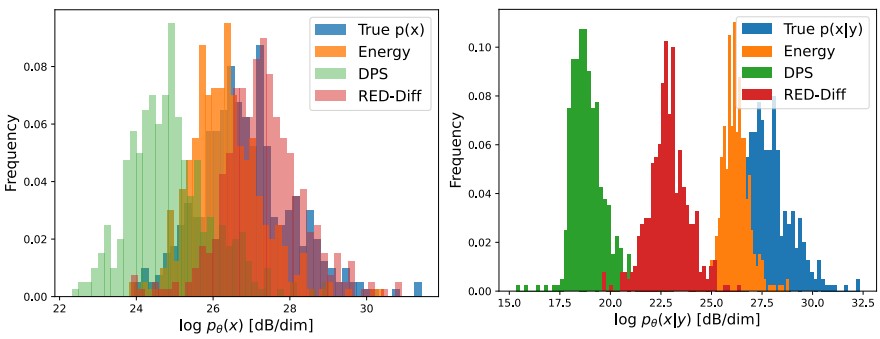

*Figure 15.* Histograms of $\log p_\theta(\hat{\mathbf{x}})$ and $\log p_\theta(\hat{\mathbf{x}}|\mathbf{y})$ for inpainting solutions $\hat{\mathbf{x}}$ generated by DPS, RED-Diff, and our energy model for different measurements $\mathbf{y}$ associated to different $\mathbf{x}$, along with the ground truths $\mathbf{x}$. Our energy model is well-calibrated with respect to both prior and posterior probabilities.

$$\log p_\theta(\mathbf{x})$$

$$\log p_\theta(\mathbf{x}|\mathbf{y})$$

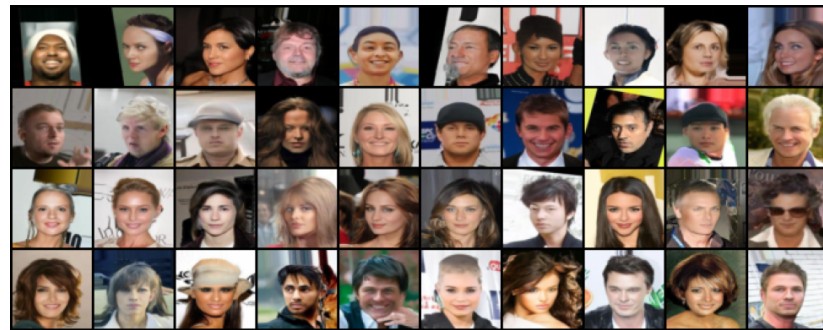

*Figure 16.* Images sorted from high to low probability (from left-up corner to right-down corner). Images with highest prior probability contains less details than those associated with higer probability.

**ImageNet.** We evaluate our model on ImageNet64, first validating that the learned prior aligns with the complexity-probability relationship described in Guth et al. (2025). As illustrated in Fig. 17, images sorted by decreasing probability transition from sparse, low-detail structures to high-complexity textures; the highest-probability samples typically feature single objects on plain backgrounds, while the lowest-probability samples are densely textured.

We extend this analysis to samples sorted by the posterior distribution $\log p(\mathbf{x}|\mathbf{y})$ in Fig. 18. We observe that the highest posterior probability does not necessarily correlate with the best reconstruction or visual quality. Furthermore, the complexity-probability trend observed in the prior persists in the posterior case.

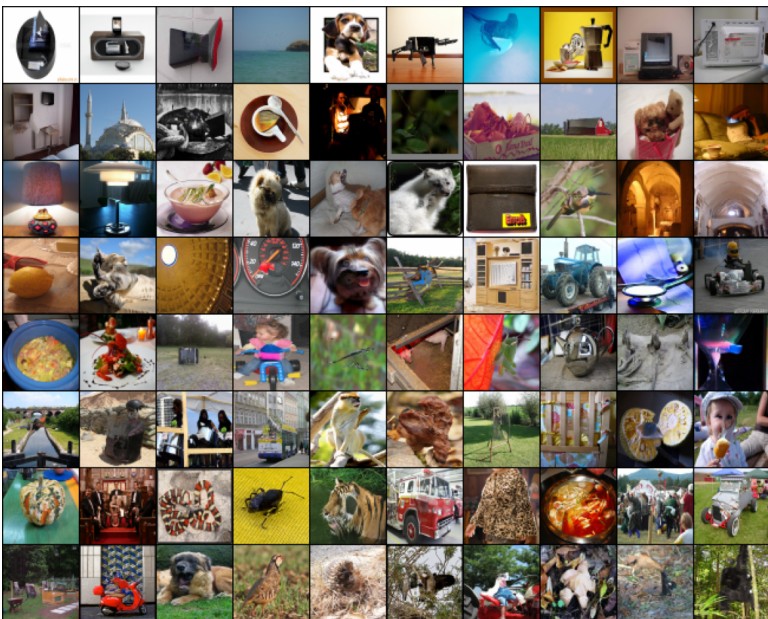

Figure 17. ImageNet posterior samples for a $21 \times 21$ center inpainting task, sorted by decreasing prior probability. Consistent with previous observations, samples with the highest probability feature uniform backgrounds and minimal detail, whereas lower-probability samples exhibit significantly higher structural complexity and dense detail.

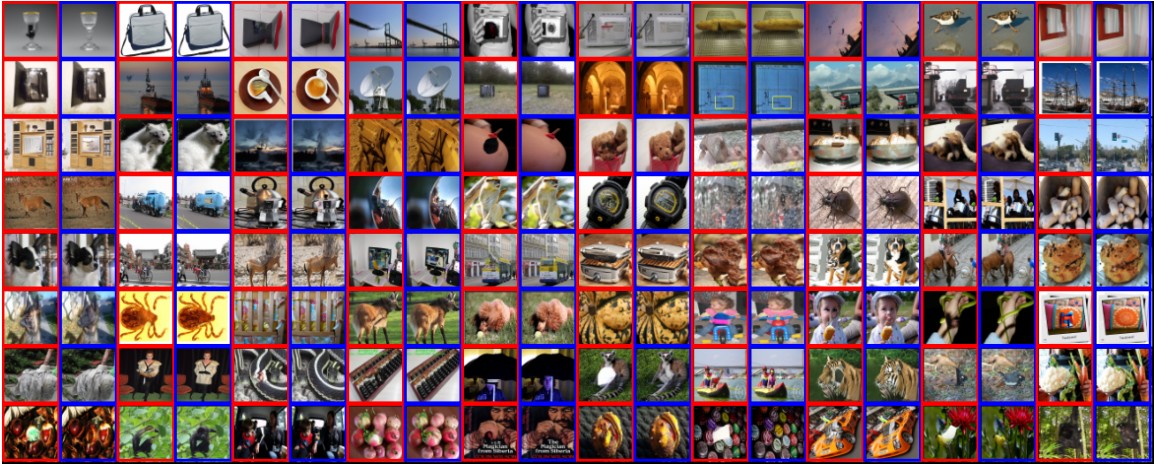

Figure 18. ImageNet posterior samples (red border) generated by sampling from measurements $\mathbf{y}$ generated using a box inpainting mask with a box of size $21 \times 21$, and sorted from high to low posterior probability, next to their corresponding ground truth (blue border).

### D.4. Recovering the Gumbel distribution for the isotropic case

Our model generalizes the energy model of (Guth et al., 2025), which is recovered as a special case by setting $\boldsymbol{\Sigma}_t = \sigma_t^2 \mathbf{I}$ (isotropic covariance). Under this reduction, the anisotropic score matching objective reduces to the standard dual score matching of [2], and the Gumbel distribution result therefore carries over directly. We include a curve computed over 50k images alongside with some of the images, which can be found in Figs. 19 and 20

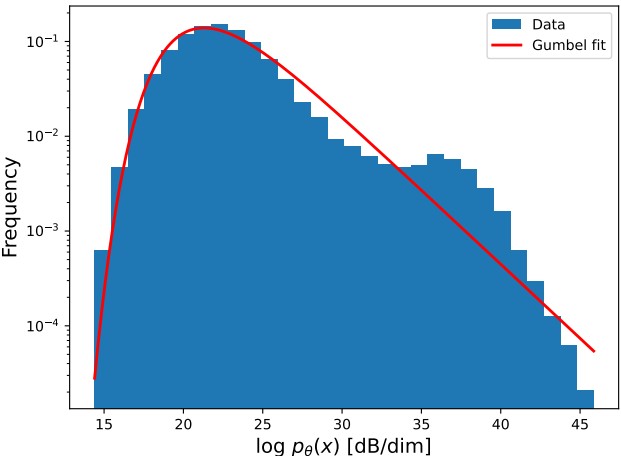

*Figure 19.* Histogram of log probabilities $\log p_\theta(\mathbf{x})[\mathrm{dB/dim}]$ for 50k in the ImageNet test set. The distribution is well-fit by a Gumbel distribution (red line), but there is a second mode (centered near 38 dB/dim) in contrast to the model in [1]. This additional mode is caused by grayscale images within ImageNet; these samples possess significantly higher probability under the model $U_\theta$ compared to standard RGB samples, creating an additional mode.

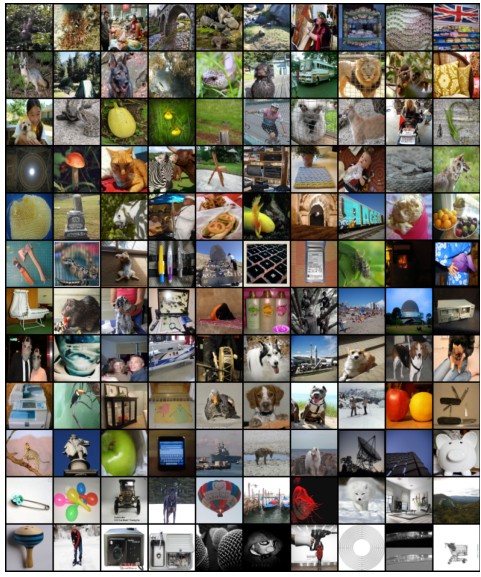

*Figure 20.* Images from the test set of ImageNet $64 \times 64$, sorted from low to high probability (left to right and top to down).

### D.5. Validation of the normalization constant in synthethic experiment

Analogous to the experiments of (Guth et al., 2025), we generate $n = 100{,}000$ samples from a mixture of two Gaussian distributions, $\frac{1}{2}\mathcal{N}(0, \sigma_1^2\mathrm{Id}) + \frac{1}{2}\mathcal{N}(0, \sigma_2^2\mathrm{Id})$, with $\sigma_1 = 1$ and $\sigma_2 = 4$, in dimension $d = 1{,}000$. Each sample receives a random partition of the $d$ dimensions into two equal groups $A$ and $B$ of size $d/2$, with two independent noise levels $t_{\mathrm{top}}, t_{\mathrm{bot}} \sim \mathrm{UniformLog}(t_{\min}, t_{\max})$, yielding per-dimension variance $\Sigma_{t,ii} = t_{\mathrm{top}}$ if $i \in A$ and $\Sigma_{t,ii} = t_{\mathrm{bot}}$ if $i \in B$. The true (normalized) energy is given by

$$U(y, \Sigma_t) = -\log\left(\sum_{i=1}^{2} e^{-\frac{1}{2}y^\top(\sigma_i^2\mathrm{Id}+\Sigma_t)^{-1}y - \frac{1}{2}\log\det\left(2\pi(\sigma_i^2\mathrm{Id}+\Sigma_t)\right)} - \log 2\right). \tag{48}$$

We parameterize the energy as a mixture of quadratics $U_\theta(y, t_{\text{top}}, t_{\text{bot}}) = -\log\Big(\sum_{i=1}^{2} e^{-a_i^{\text{top}}(t) r_{\text{top}}^2 - a_i^{\text{bot}}(t) r_{\text{bot}}^2 - b_i(t)}\Big)$, where $r_{\text{top}}^2 = \|y_A\|^2$ and $r_{\text{bot}}^2 = \|y_B\|^2$ are the squared norms over each group, and the functions $a_i^{\text{top}}, a_i^{\text{bot}}, b_i$ are computed by a 5-layer MLP with a hidden dimension of 256 that takes the 2-dimensional schedule embedding $[\log(t_{\text{top}} + t_{\text{min}}), \log(t_{\text{bot}} + t_{\text{min}})]$ as input (reducing to the isotropic case when $t_{\text{top}} = t_{\text{bot}}$). This network is trained either with single (space) score matching or with dual score matching, both across noise levels $t_{\text{top}}, t_{\text{bot}} \in [t_{\text{min}}, t_{\text{max}}]$, where the dual loss includes one time-derivative term per noise level. Training is otherwise similar to the isotropic experiment, for 50,000 training steps with a batch size of 512 and an initial learning rate of 0.0001, over noise levels from $t_{\text{min}} = 10^{-2}$ to $t_{\text{max}} = 10^2$. The comparison is illustrated in Fig. 21

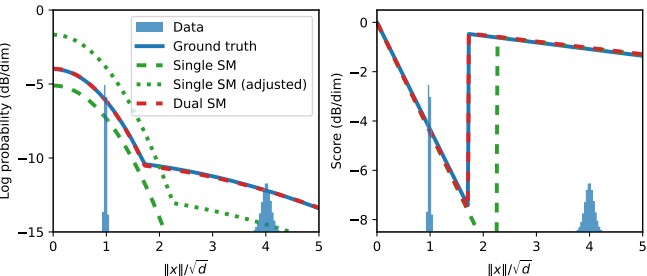

*Figure 21.* Comparison of single and dual score matching on a 1000-dimensional Gaussian scale mixture. Left: Dual score matching (red dashed) captures the true energy (blue solid), whereas single score matching (green) fails, even after global normalization. Right: Radial score components. Single score matching learns the score accurately within the data support (blue bar plot) but lacks accuracy outside of this region.

## D.6. Reconstructed images

| Measurement | Ours | DPS | RED-Diff | DAPS | Ground-truth |
| --- | --- | --- | --- | --- | --- |

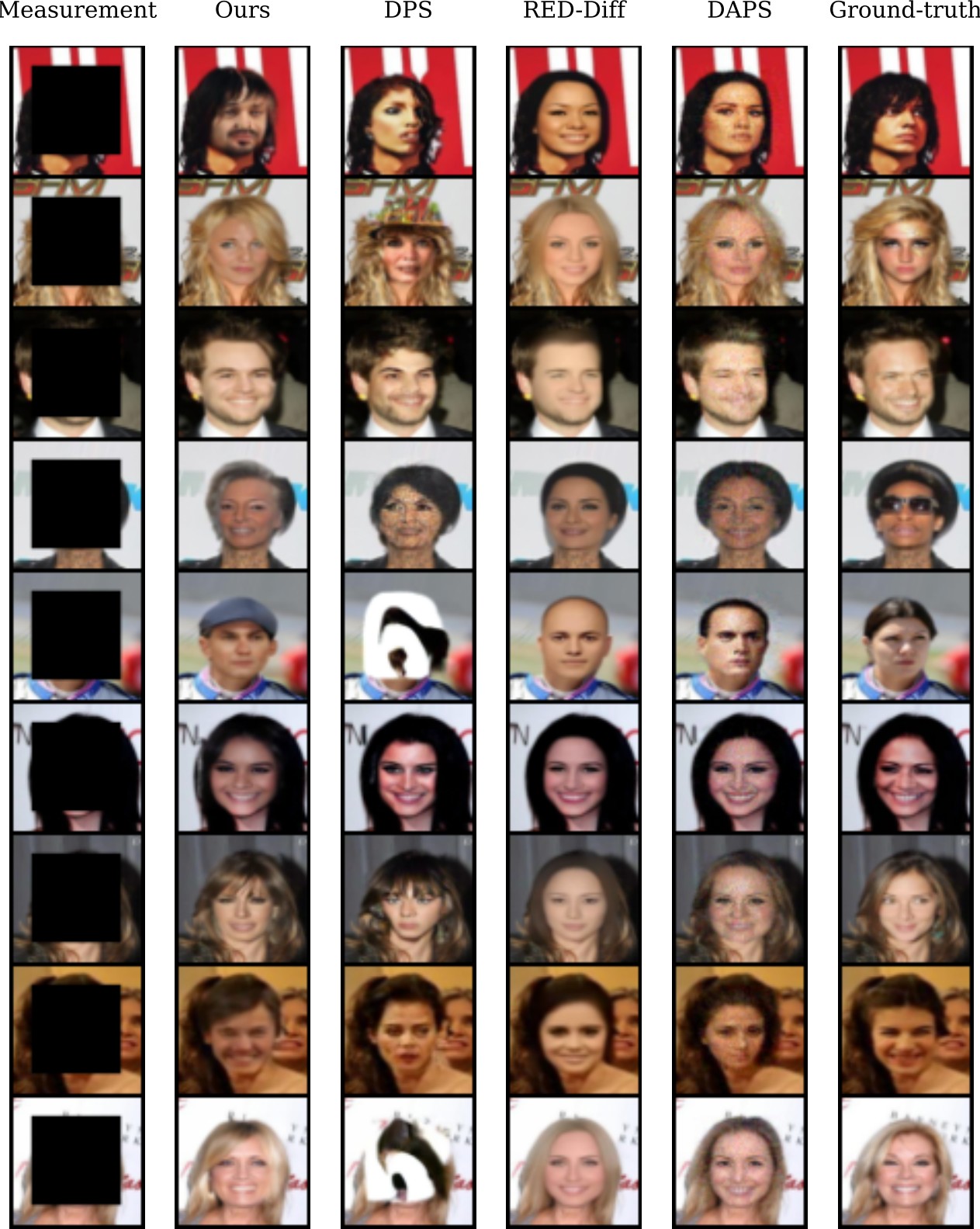

*Figure 22.* Examples of inpainting for CelebA.

| Measurement | Ours | DPS | RED-Diff | DAPS | Ground-truth |
|---|---|---|---|---|---|

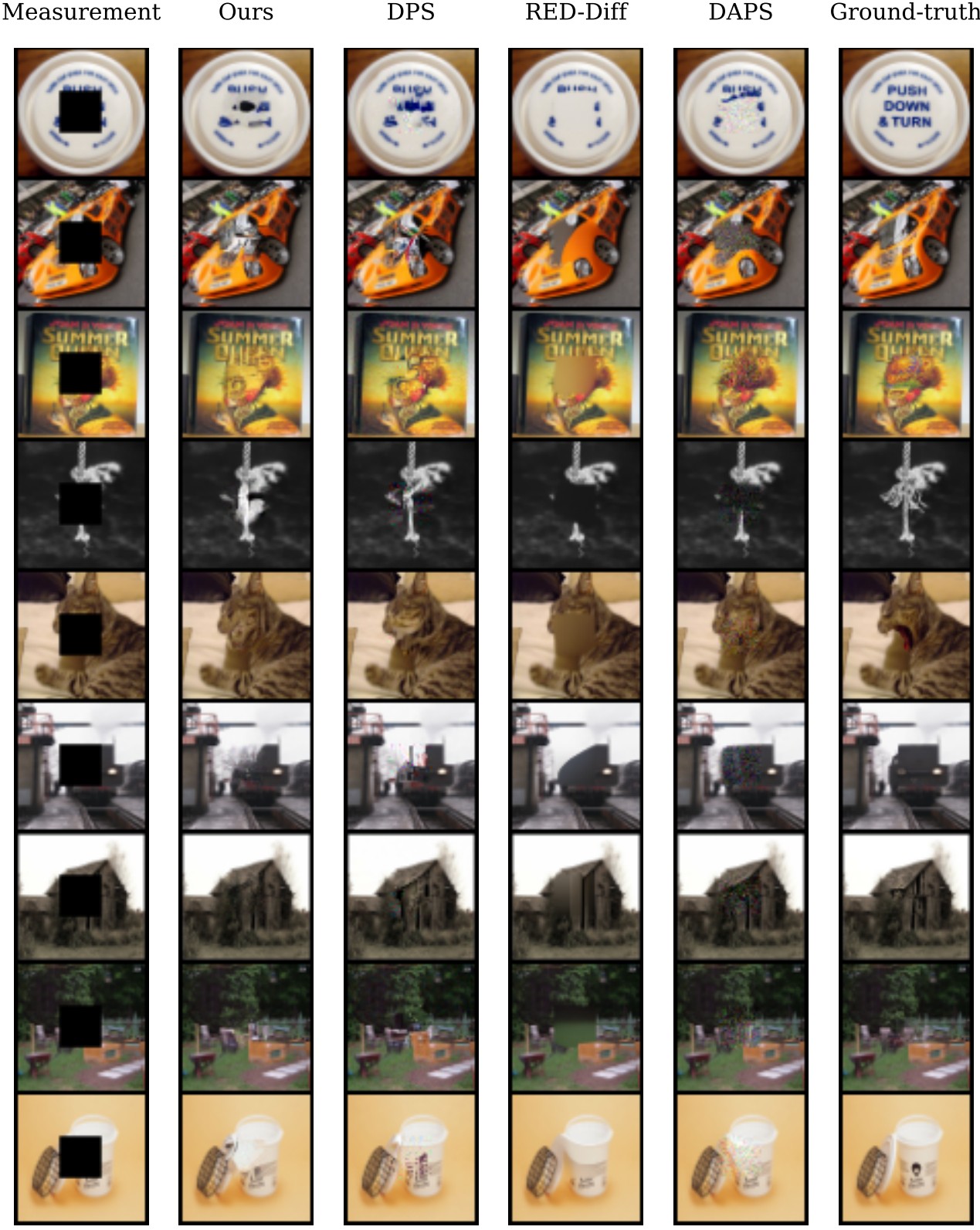

*Figure 23.* Examples of inpainting for ImageNet.

| Measurement | Ours | DPS | RED-Diff | DAPS | Ground-truth |
|---|---|---|---|---|---|

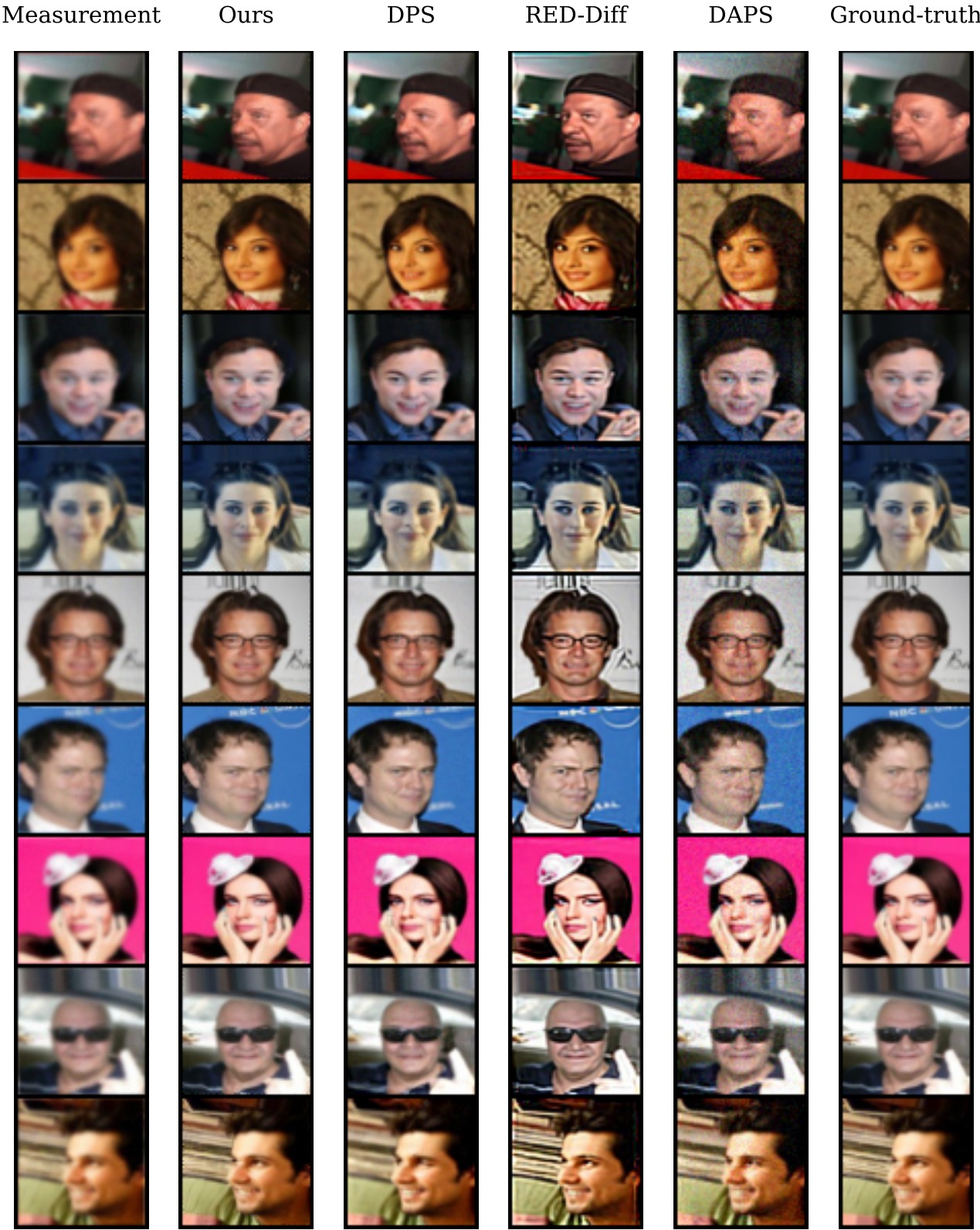

*Figure 24.* Examples of deblurring for CelebA

Measurement  Ours   DPS   RED-Diff   DAPS Ground-truth

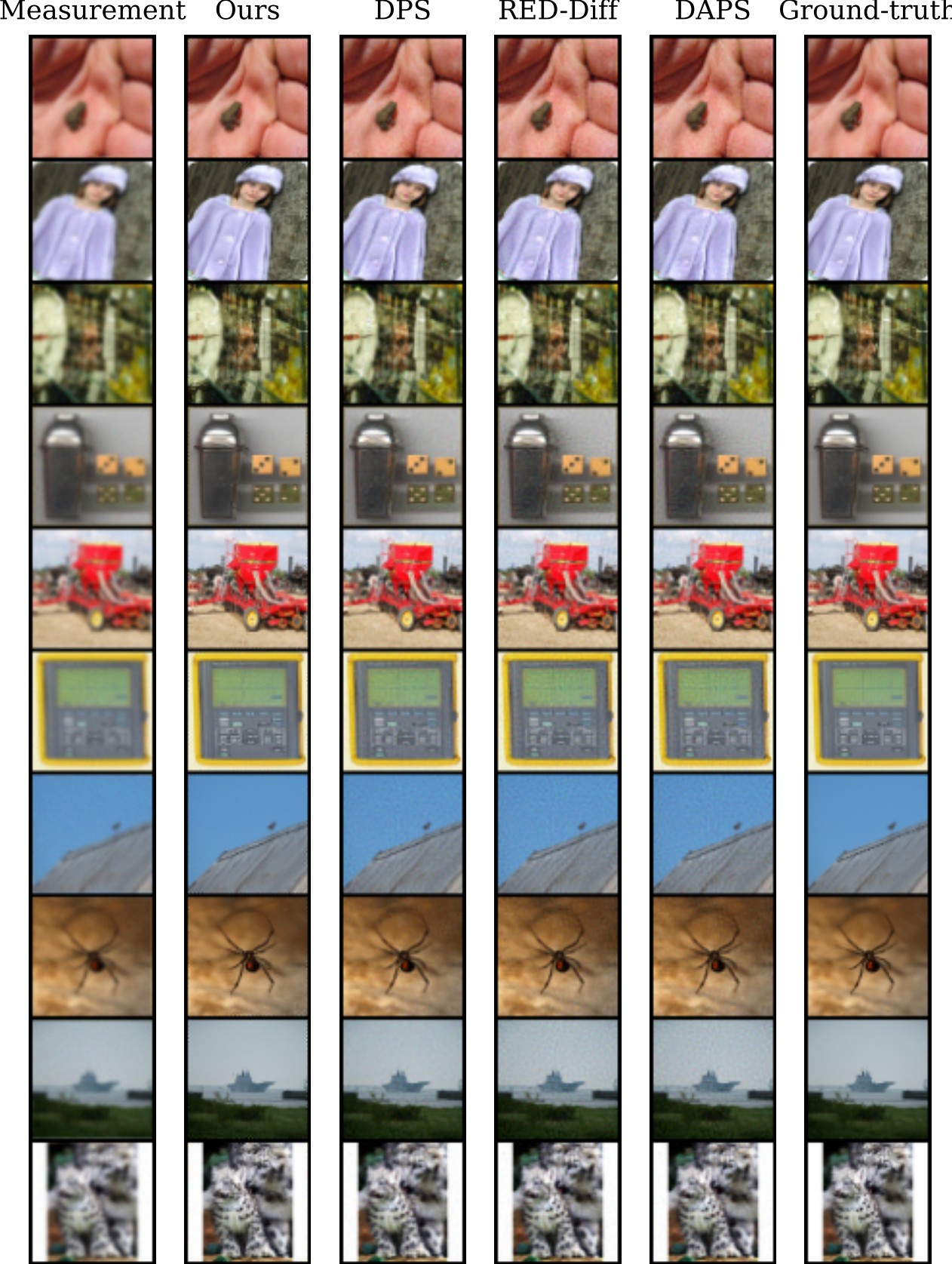

*Figure 25.* Examples of deblurring for ImageNet

Measurement          Ours          DPS          RED-Diff          Ground-truth

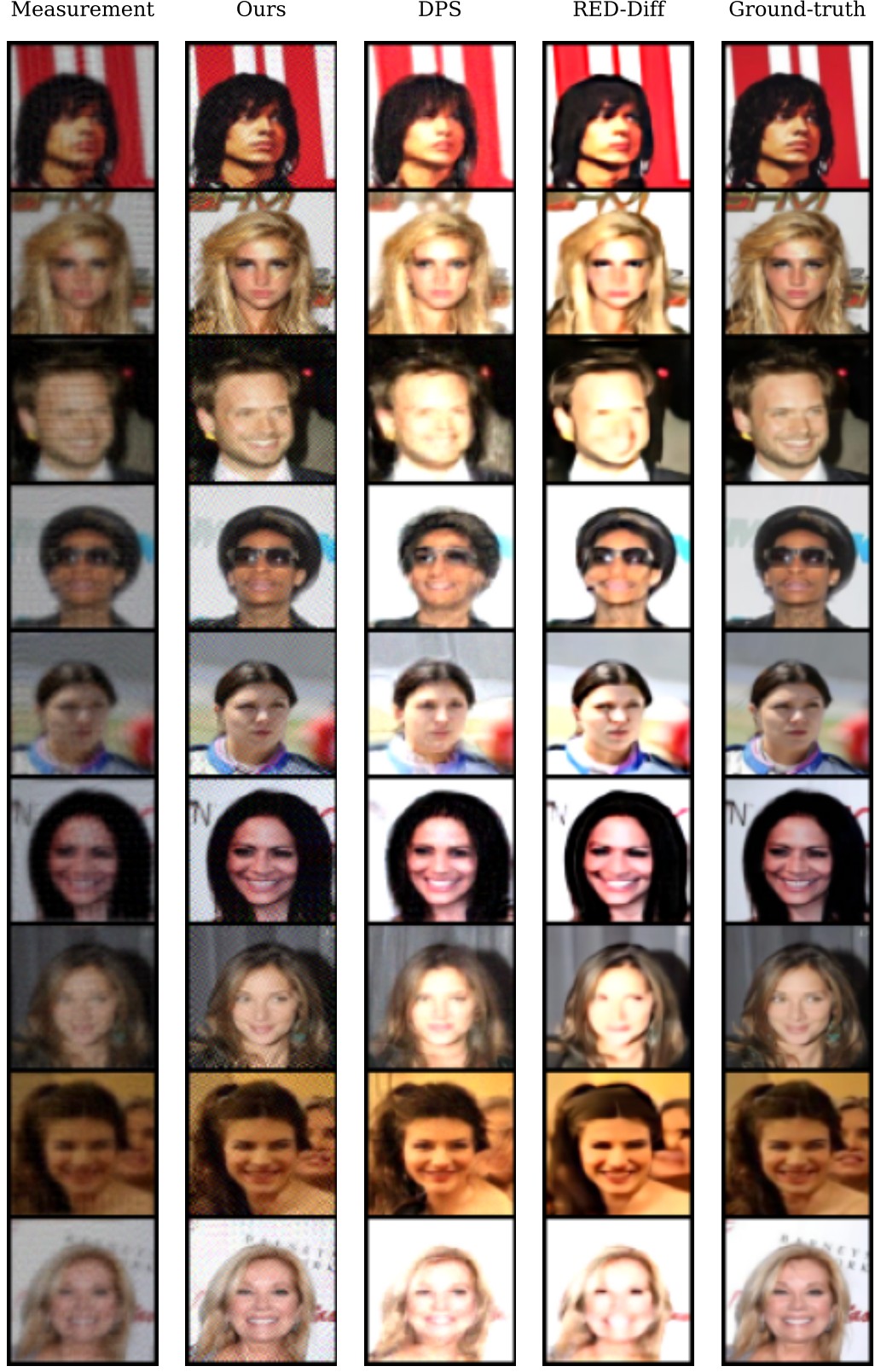

*Figure 26.* Examples of super-resolution for CelebA

Measurement          Ours          DPS          RED-Diff          Ground-truth

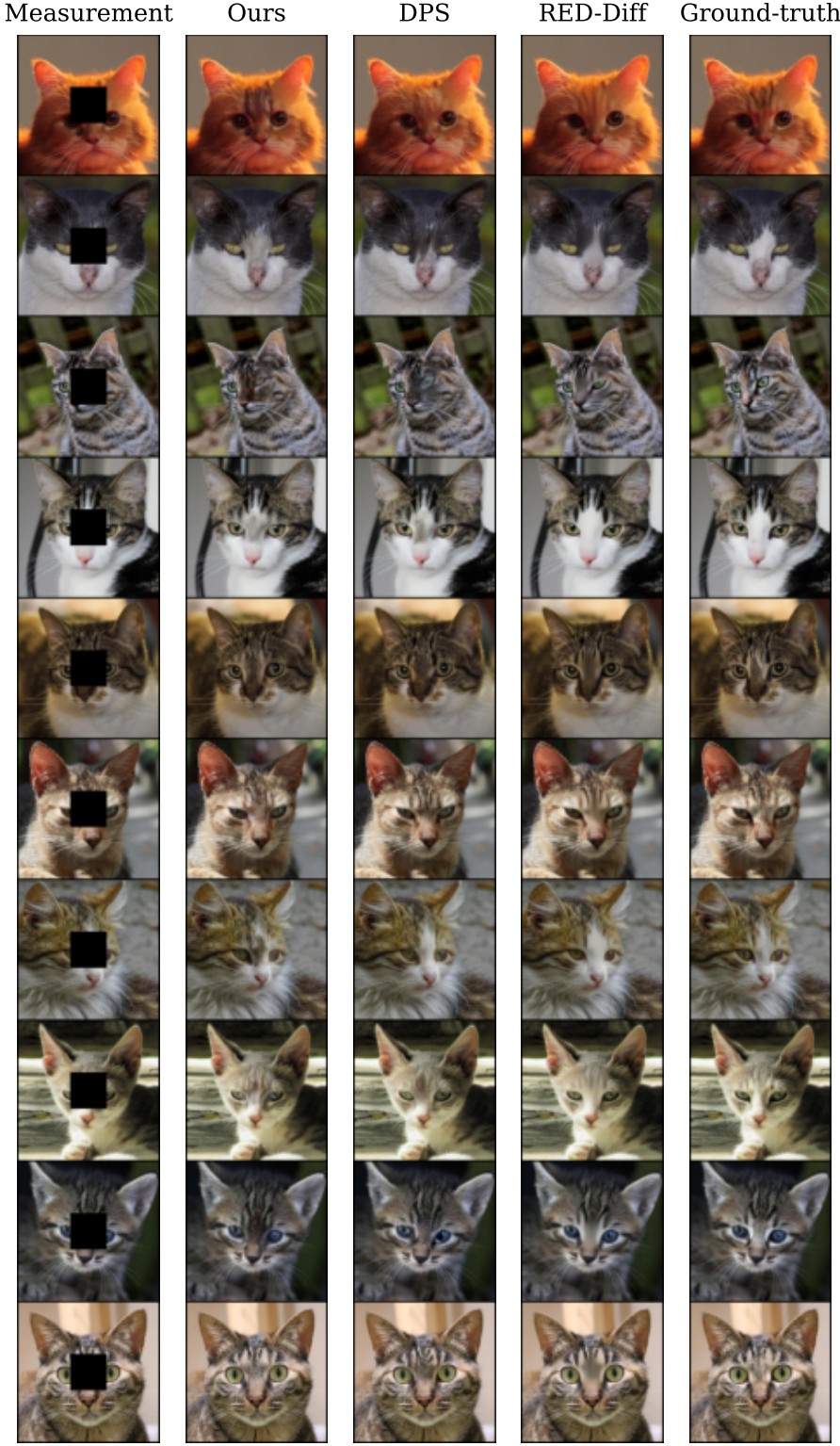

*Figure 27.* Examples of inpainting for AFHQ-cats.

Measurement   Ours    DPS   RED-Diff  Ground-truth

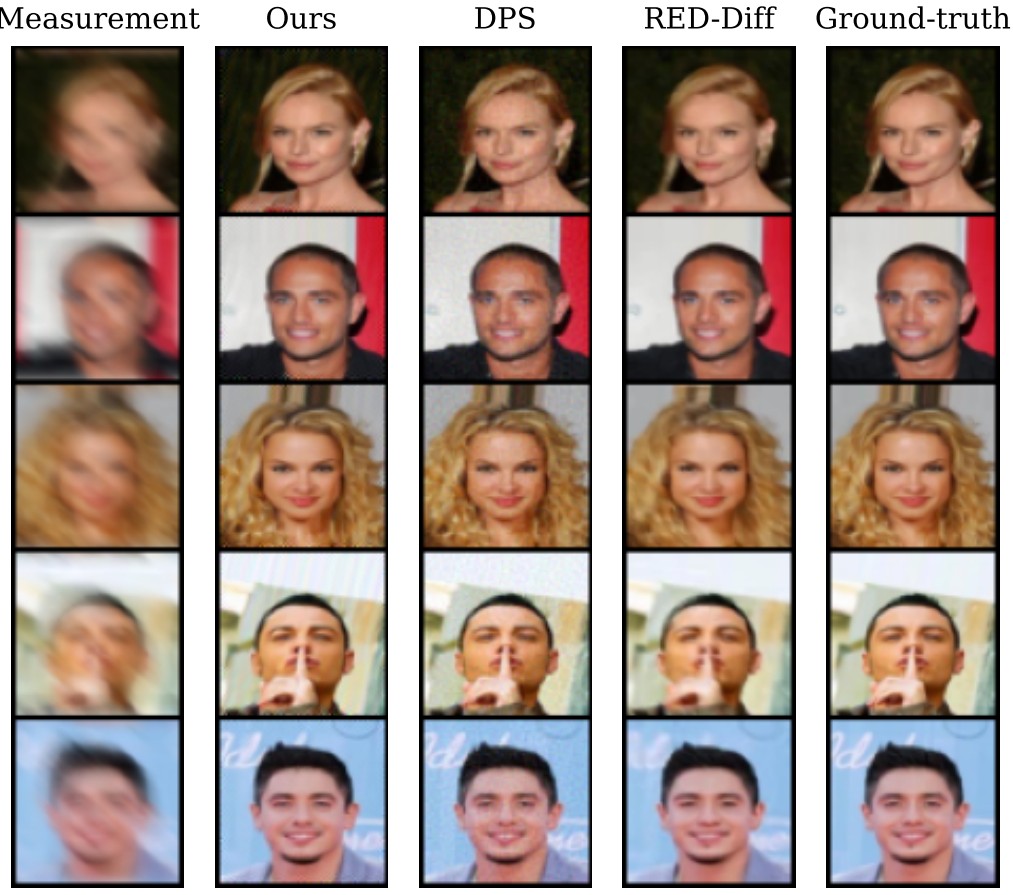

*Figure 28.* Examples of Motion Deblurring.

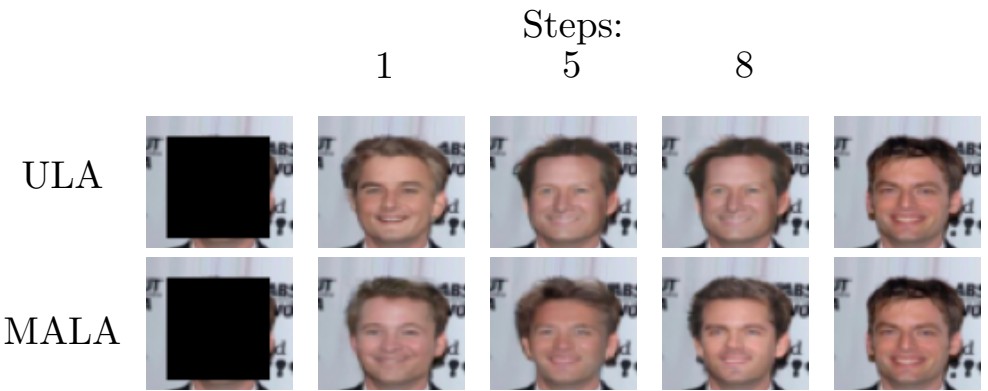

*Figure 29.* Comparison of the different corrector samplers a function of the number of steps.

