# OpenReview forum: "Learning Normalized Energy Models for Linear Inverse Problems"
_ICML.cc/2026/Conference — ICML 2026 regular_

### Official Review · Reviewer_mbr1 · 2026-03-04

**Soundness:** 3
**Presentation:** 3
**Significance:** 2
**Originality:** 3
**Overall Recommendation:** 3
**Confidence:** 5

**Summary:**

This paper proposes a new strategy to learn an energy model for linear inverse problem solved with diffusion models. It develops a method which takes advantage of anisotropic noise with learning $-\log p(y|\Sigma)$ with $y$ the observation and $\Sigma$ the covariance matrix which encoded the noise level and the forward operator of the problem.

**Compliance With Llm Reviewing Policy:**

Affirmed.

**Key Questions For Authors:**

- About the reference [1]: I do not see in this reference where the authors precise that they use the parametrization described in equation (11) for the energy model.
- Paragraph 3.3(ii) : If the covariance can be diagonal in the spatial or frequency domain, is-it not $2d$ parameters of freedom instead of $d$ ?
- Paragraph 3.3(iii) : Why using the SiLU activation fonction ?
- Images in experiments are relatively small (64*64), why this choice ? What is the computational cost of your method compare to others ? Please add it in Table 1.
- How do you compute experimentally the gradient of $U_{\theta}$ with respect with $y$ ?
- In Table 1, I am surprised of the large gap of performances between your method and the others for Deblurring with CelebA. How do you explain that ?
- In Figure 19/20, it seems that your method introduce some artifacts on the border of the image that does not appear in other methods or ground-truth. How do you explain this ?
- In order to compute the likelihood in Figure 4, do you use your own learned energy based model for every method ? If it is not the case, how do you compute it for DPS and RED-Diff ? How many samples have been generated for Figure 4 ?
- Figure 12 left, how do you know the true $p(x|y)$ histogram in this situation ?
- Figure 13 up, what is the observation $y$ use for this generation ?
- Figure 15, could you detail more how this figure is constructed ? What are the observation involves ? How do you know exact posterior sample (in red) ?
- Could you use the energy-based model to compute $p(y)$ and discrimate a priori between probable and unprabable observations ? It could give an interesting information if the restoration task is "in-distribution" or "out-distribution".
- In equation (19), there is an inversion between the gradient and the expectation. Under what conditions this inversion is valid ?
- Do you recover the Gumbel distribution as described in [2] with your learned prior ?

[1] Composition and control with distilled energy diffusion models and sequential monte carlo, Thornton, James and B{\'e}thune, Louis and Zhang, Ruixiang and Bradley, Arwen and Nakkiran, Preetum and Zhai, Shuangfei, Int. Conf. on Artif. Intell. and Stat., 2025
[2] Learning normalized image densities via dual score matching, Guth, Florentin and Kadkhodaie, Zahra and Simoncelli, Eero P, Neurips 2025

**Limitations:**

Yes the limitations are clearly discussed.

**Strengths And Weaknesses:**

Strength:
- The idea of the paper is original and is part of a recent trend to design energy-based model. It is very interesting for image restoration as it allows to evaluate the learned prior bridging the gap between classical method with explicit prior and modern learned implicit priors.
- The experiments are developped in details with many examples in Appendices.
- The code is provided.

Major Weaknesses:
- The way of choosing the learned covariance (diagonal in pixels or frenquencies space) seems a bit arbitral. Moreover, the model is not tested on experimental setting far from these hypothesis (blur with non-periodic bound conditions or non-Gaussian noise for instance).
- Even if, it is mentionned in the limitations, clear quantification of the computational cost of training and sampling of the method are not provided. It is only stated to be slower than others.
- The restriction on the inverse problem tackable is larger than stated :
           - It focuses on linear inverse problem with additive Gaussian noise, as detailed in equation (4) without being mention explicitely in the abstract and the introduction.
           - It only focuses on symetric $H$. It is not exact that equation (4) is equivalent to equation (6) even in the case chere $H$ is invertible. It is also needed that $H$ is symetric which is not precise here and is restrictive (exclude non-symetric kernel of blur such as motion blur).

Minor Weaknesses/Typos:
- Between formula (1) and (2), please adopt the same convention for the noise. It seems a bit unsual to me to add the normal density to the process. Same remark in equation (12).
- line 199, it seems in the Appendices that there is a weigthed parameter between the two terms of the loss, please mention it in the main paper.
- line 204, it will be more correct to write $p(\cdot|\Sigma) = \mathcal{N}(0,\Sigma)$ or $p(y|\Sigma) = \mathcal{N}(y; 0,\Sigma)$. In the current formulation it is confusing, the density is evaluated at $y$ on one side of the equation but not on the other.
- equation (10), I do not understand why there is the constant $e$ in the determinant, for me it should be $\log \text{det}(2\pi\Sigma)$.
- Equation (11) : Please refer to [1] that is the first paper to propose this parametrization in 2017.
- Line 205 left : Please give a reference for the SiLU activation fonction and give its formula, this is not standard.
- In Figure 17-18-19-20-21, could you put some metric with the images (PSNR, LPIPS) to have some quantitative numbers with these qualitative tables ?
- Equation (13) is not really motivated in the main paper.
- Line 642, What is "NLL" ?

The code is not reproductible:
- The README.md file is not complete. In particular, it is hard to understand the structure of the code.
- The pre-trained checkpoints given in the README.md file are not available.
- There is no file with the exact library setting to use for running the code. This is crucial for reproductibility since the libraries evolve rapidly.
- It seems to have a lot of useless files in the code making it a bit confusing.

[1] The little engine that could: Regularization by denoising (RED), Romano, Yaniv and Elad, Michael and Milanfar, Peyman, SIAM journal on imaging sciences, 2017

---

> ### Author Rebuttal · Authors · 2026-03-31
>
> Thank you for the detailed review! We address your concerns below; Figs. and a README for reproducibility are in :https://anonymous.4open.science/r/rebuttal-211F/rebuttal_images.pdf. Due to space limitations, we point to other reviewers' responses for similar questions: for scalability we deferred to W2 from Reviewer 1; for the gumbel distr. to W4 from Reviewer 2, and for the diagonal desing choice to W2 from Reviewer 2.
> # W1 Choice of $\Sigma$ and problem restriction
> 1. We agree the Gaussian noise restriction should be stated explicitly, we will clarify the abstract and introduction. We believe our formulation extends to other noise types (e.g., Poisson), where equivalent relations between energy derivatives and optimal denoisers hold [1-2].
> 2. We respectfully disagree that the equivalence between Eq.(4)-(6) requires $H$ to be symmetric: the distribution of $\sigma H^{-1}v$ equals $\Sigma^{1/2}v$ when $v\sim\mathcal{N}(0, I)$, so non-symmetric blur kernels are handled. However, like other methods focusing on linear measurements with additive Gaussian noise, complex degradations like motion blur are outside our scope.
> # W2 Comp. cost
> - For sampling, the additional $\nabla_yU_\theta$ gradient is similar in cost to DPS's backpropagation and architectural overhead is minimal. Thus, wall-clock time is approx. equal at the same NFEs. For AFHQ $192\times192$: DPS 55sec, ours 74sec.
> - Training requires approx. double the memory due to double backpropagation and the dual loss. On CelebA $64\times64$: score model takes $\sim$ 2 days (4 GPUs), ours $\sim$ 11 days (6 GPUs).
> # W3 Notation
> We agree some notation is unclear, and will unify: using $v$ for Gaussian noise, define NLL as negative log-likelihood, and clarify the weighting in $\ell_{A-CSM}$ (norm weighting in eq.(25) embeds it).
> # W4 Constant $e$
> The expression is correct. For $y \sim \mathcal{N}(0, \Sigma_T)$, the differential entropy is $H = \frac{1}{2}\log \det (2\pi e\Sigma_T)$; the extra $e$ arises from the quadratic term $\frac{1}{2}\mathbb{E}[\mathrm{tr}(\Sigma_T^{-1}yy^\top)] = \frac{d}{2}$.
> # W5 SiLU
> We will add the corresponding reference as well as the definition $\text{SiLU}(x) = x\cdot\text{sigmoid}(x)$. We use SiLU to align with [Karras 2022] and because it outperforms all tested activations, besides being differentiable w.r.t. $y$, unlike ReLU.
> # W6 Metrics for Figs.17-21
> These are samples from Sec.3.4 experiments, whose metrics are in Tab. 1. We agree this is unclear and will add per-image metrics.
> # W7 Eq.(13)
> The motivation is provided in App.B due to space constraints.
> # Q1 Ref. of eq.(11)
> We will add [1RED]. Please, note eq.(11) in [1Thorn.], $F_\theta(x_t)=h_\theta(x_t)\cdot x_t$, corresponds to our inner product parametrization up to a $\frac{1}{2}$ factor.
> # Q2 Degrees of freedom
> Please, notice $\Sigma$ is defined in a single basis (spatial or frequency), reducing from $\frac{d(d{-}1)}{2}$ to $d$.
> # Q4 Deblur. gap
> Guidance-based methods approximate the posterior by combining a prior with a likelihood at inference time, which degrades for anisotropic $H$. Our model learns $p(x|y)$ directly with the correct covariance structure (FFT of blur kernel). We emphasize hyperparameters were tuned per method (see W2, Reviewer Tgke).
> # Q5 Border artifacts
> For simplicity,  we use periodic boundary conditions, which can cause border effects; a DCT or more advanced methods should improve.
> # Q6 Figs. clarifications
> - Fig4 uses $\log p(x|y)\approx\log p(y|x)-U_\theta(x)+U_\theta(y)$, with closed-form likelihood $\mathcal{N}(y;Hx,\sigma_v^2I)$ applied identically across methods; priors are approximated with $U_\theta(.)$ using 400 samples using different seeds.
> - Fig12: The histogram evaluates $U_\theta$ on ground-truth images, serving as a reference under our model, not the true analytical posterior.
> - Fig13: $y$ is from the inp. task in Sec.3.4 (masked observation from Fig.17).
> - Fig15 are posterior samples for $21\times21$ box inpainting, where each test img. $x$ is corrupted to get $y$, and sorted by $U_\theta(\hat{x}|y)$.
> # Q7 $U_\theta$ as discriminator
> Unfortunately, OOD images do not necessarily have low probability under the model: [Guth 2025] shows blurring increases likelihood, and [3] finds high-density regions correspond to unrealistic, cartoon-like images.
> # Q8 Eq.(19)
> Please, notice this is not an inversion. In particular:
> $U_\theta(y, \Sigma) = -\log \int p(x_0) p(y|x_0, \Sigma) dx_0$. Taking $\nabla_\Sigma$ and using the log-derivative trick, we get $U_\theta(y, \Sigma) = -\frac{\int p(x_0) p(y|x_0, \Sigma) \nabla_\Sigma \log p(y|x_0, \Sigma) dx_0}{\int p(x_0) p(y|x_0, \Sigma) dx_0}$ which is exactly $\mathbb{E}_{x_0}[-\nabla\_\Sigma \log p(y|x_0,\Sigma)|y,\Sigma]$.
>
> [1]Bhattacharya,S., et al.ItDPDM:Information-Theoretic Discrete Poisson Diffusion Model.
> [2]Raphan, M., et al. Least squares estimation without priors or supervision.
> [3]Karczewski, R., et al.Diffusion Models as Cartoonists: The Curious Case of High Density Regions.

---

> > ### Author Rebuttal · Reviewer_mbr1 · 2026-04-02
> >
> > Thank you for your detailed response.
> >
> > I apologize for my confusion on symetric $H$. I understand that you only claim that the distribution of $\sigma H^{-1}v$ and $\Sigma^{\frac{1}{2}}v$ are the same. I was thinking that you claimed that $\sigma H^{-1} = \Sigma^{\frac{1}{2}}$ which is not true in general (if $H$ is not symetric in particular). Thanks for your explanation !
> >
> > I thank the authors for the correction in the code. It seems to be more reproducible. I do not review the second version in details but the requierements.txt, a clear README.md file are provided. Please make sure that the weights will be available for the readers (the current link is only for a week).
> >
> > My first main weakness on the choice of learned covariance is not answer. And I do not see where a clear comparison in term of computational cost is provided between the different methods. Since two other three of my main weaknesses still hold, I keep my score.

---

> > > ### Author Response · Authors · 2026-04-03
> > >
> > > We truly thank the reviewer for engaging with the discussion and are pleased that we were able to address several concerns. We will make sure to have the weights available for readers, and we will try to resolve the remaining ones below.
> > >
> > > # Computational cost
> > > We expand on the running times for Gaussian deblurring across all methods, considering the running time for sampling a batch of 16 images at 64x64
> > >
> > > |Sampler|Time [sec]|Computational cost (NFE)|
> > > |:-|:-:|:-:|
> > > |Ours|139|$(N_p)$ NFEs + $(N_p)$ grad w.r.t. input for energy|
> > > |DPS|167|$N$ NFEs + $N$ grad w.r.t. denoiser|
> > > |RED-Diff|22|$N$ NFEs|
> > >
> > > Our method sits between DPS and RED-Diff in computational cost, while being the only one that learns a full posterior density. We believe this makes the overhead modest relative to the added capabilities our approach unlocks, such as blind estimation and normalization constant computation, which are not possible with DPS or RED-Diff.
> > >
> > > # Additional non-symmetric degradations
> > > First, the diagonal constraint is a design choice driven by computational tractability: a general $\Sigma$ would require $\frac{d(d-1)}{2}$ parameters (mentioned in Section 3.3, item $(ii)$), which is prohibitive even for images at $64\times64$. This assumption aligns with recent work using an anisotropic process.
> > >
> > > Thus, two natural choices arise for $\Sigma$: 1) diagonal in the pixel domain, which covers inpainting and spatially varying noise; and 2) diagonal in the frequency domain, which covers Gaussian deblurring and super-resolution. We emphasize that a broad class of linear inverse problems considered in prior works falls in the union of these two categories. Moreover, we can handle **both** categories with a **single** network, a strength and unique contribution of our proposed methodology that goes beyond previous works.
> > >
> > > To further illustrate that our model can handle a non-symmetric blur kernel, we include an additional experiment on Motion deblurring (please see our updated PDF, Fig.8 for visual results) with $\sigma_v = 2\times10^{-3}$ (noise variance).
> > >
> > > |Sampler|PSNR [dB] $\uparrow$|LPIPS $\downarrow$|FID $\downarrow$|DISTS $\downarrow$|
> > > |:-|:-:|:-:|:-:|:-:|
> > > |**Ours**|29.42|**0.007**|50.6|**0.083**|
> > > |DPS|28.63|0.02|45.65|0.094|
> > > |RED-Diff|**31.00**|0.02|**44.33**|0.085|
> > >
> > > As the numerical results illustrate, our method is competitive in this OOD covariance. In particular, given that the anisotropic scheduling yields a true posterior sampler, it achieves the best LPIPS across methods. However, the FID is higher than baselines, likely due to border artifacts from periodic boundary conditions in the Fourier transform (as discussed in Q5).
> > >
> > > We remark that this deblurring type is **not** part of the training set, demonstrating that our model generalizes to degradations not seen during training. We will include this new experiment (Motion Deblurring) in the revised manuscript and welcome specific suggestions from the reviewer to enhance our work.
> > >
> > > # Bug in Gaussian deblurring
> > > We would like to thank the reviewer for pointing out additional degradations and for noticing the gap between the baselines and our method for Gaussian deblurring. We found a bug in our evaluation (we were using H^2 instead of H for the Gaussian deblurring likelihood term in guidance-based), which we have now corrected. The updated results are more consistent with expectations: our method achieves the best perceptual quality (LPIPS, FID, DISTS), while being slightly below DPS and RED-Diff in terms of PSNR. This aligns with the perception-distortion trade-off analysis, supporting our claim that the anisotropic scheduling yields a better posterior sampler. We apologize for this confusion and will update the table in the revised manuscript.
> > >
> > > |Sampler|PSNR [dB] $\uparrow$|LPIPS $\downarrow$|FID $\downarrow$|DISTS $\downarrow$|
> > > |:-|:-:|:-:|:-:|:-:|
> > > |**Ours**|32.27|**0.002**|**41.8**|**0.04**|
> > > |DPS|32.98|0.004|43.01|0.08|
> > > |RED-Diff|**34.53**|0.006|44.33|0.08|
> > >
> > > We emphasize that these corrected results do not affect the main contribution of our paper, which is not building a SOTA denoiser, but rather learning a posterior density that unlocks several new capabilities beyond standard reconstruction.
> > >
> > > # Closing remark
> > > We respectfully note that while the reviewer's concerns about covariance and computational cost are valid concerns, we believe the evidence presented (OOD generalization to motion deblurring, competitive perceptual metrics, and unique capabilities such as blind estimation and normalization constant computation) demonstrates that the diagonal assumption is effective, and the choice aligns with the majority of papers in the literature. Our paper's contribution lies not in achieving SOTA reconstruction on individual tasks, but in providing a principled framework for learning posterior densities over a family of linear inverse problems with a single model. We hope the reviewer will consider these strengths, and the additional experiments.

---

### Official Review · Reviewer_937Z · 2026-03-10

**Soundness:** 3
**Presentation:** 4
**Significance:** 4
**Originality:** 3
**Overall Recommendation:** 5
**Confidence:** 3

**Summary:**

This paper proposes a normalized EBM trained via anisotropic covariance score matching to provide an explicit posterior for the linear inverse problem. The method features a novel covariance-conditioned U-Net architecture compatible with both spatial and frequency domain kernels, and it enables adaptive energy-guided samples, unbiased MCMC corrections, and blind degradation estimation. Experiments on multiple tasks demonstrate competitive performance.

**Compliance With Llm Reviewing Policy:**

Affirmed.

**Final Justification:**

The rebuttal has solved most of my concerns, and I tend to accept it.

**Key Questions For Authors:**

Please see in weakness.

**Limitations:**

Yes

**Strengths And Weaknesses:**

Strength: This paper presents a significant technical advancement by extending dual-score matching to A-CSM, effectively bridging the gap in handling structured, correlated noise for diffusion-based EBMS. The proposed framework enables the learning of unified, normalized, and explicit posterior densities within a single model.
Comprehensive quantitative results and high-quality visual analyses across the main text and appendix demonstrate that the proposed method consistently outperforms the baseline in inverse problems (e.g., deblurring, inpainting, and SR).

Weakness:
1. In section 3.3, the authors restrict the model to diagonal covariance structures in either spatial or frequency domains to reduce the parameter count from $\frac{d(d-1)}{2}$ to $d$. This oversimplified assumption implies that the framework is incapable of modeling non-uniform blur or other complex noises. While the authors briefly acknowledge this in the discussion, the paper lacks an evaluation of how this simplification affects performance in non-idealized wetting. Providing quantitative evidence for both variants is essential to justify this design choice.
2. The experiment is significantly constrained by the low resolution of the datasets used, specifically CelebA (with 64 \times 64 resolution). While the author acknowledges that the proposed method is computational intensive, the reliance on such toy-scale images makes it difficult to assess the model’s efficacy and scalability for real-world inverse problems, which typically involve resolutions of 256 \ times 256. Without experimentation of the model’s performance on higher-dimensional data, the practical impact and robustness of the proposed method remain unverified.

---

> ### Author Rebuttal · Authors · 2026-03-31
>
> Thanks for your positive feedback on our paper, and your review! We ran more experiments to further improve the results and address your concerns. Link to anonymous PDF with Figures in https://anonymous.4open.science/r/rebuttal-211F/rebuttal_images.pdf
>
> # W1: Effects of simplifying the covariance and how this affect the model
> We thank you for raising this point. First, we would like to clarify that our mathematical framework for $H^{-1}$ is fully general and imposes no condition on the degradation. For cases where the inverse has infinite variance (when $H$ is not invertible, e.g., for inpainting), we can approximate those values with a large variance, as is done in diffusion models.
>
> Based on this general formulation, we enforce the diagonal constraint, a design choice driven by computational tractability: a general $\Sigma$ would require $\frac{d(d-1)}{2}$ parameters (mentioned in Section 3.3 item $(ii)$), which is prohibitive even for images at $64\times64$. This assumption aligns with recent works using an anisotropic process [1-4].
>
> Thus, two natural choices arise for $\Sigma$: 1) diagonal in the pixel domain, which covers inpainting and spatially varying noise; and 2) diagonal in the frequency domain, which covers Gaussian deblurring and super-resolution. We emphasize that a broad class of linear inverse problems considered in prior works falls in the union of these two categories. In addition, we can handle _both_ categories with a _single_ network, a strength and unique contribution of our proposed methodology that goes beyond previous works.
>
> Nevertheless, we believe there are alternative ways to extend this approximation by considering low-dimensional parameterizations of $H$, as in [5]. This is the case of several linear inverse problems in practice, where one considers a family of covariances $\Sigma_\phi$ for a low-dimensional parameter $\phi$. This can be accommodated in our framework by letting the network compute $U_\theta(y, \phi) \approx U(y, \Sigma_\phi)$. This requires modifying the embedding network and the A-CSM objective to handle this low-dimensional parameterization $\phi$.
>
> # W2 Scalability
> Scalability to high resolution
> We thank the reviewer for raising this point. Since our submission, we trained a model on the higher-resolution dataset AFHQ-cats $192\times192$ (following [1]).
> - Sampling: we compare with DPS using 800 NFEs for box inpainting; the results are shown in the following tables, and a subset of examples are in Fig.2 and 3 in the attached link.
>
> Table 1: $30 \times 30$ box
> |Sampler|PSNR [dB] $\uparrow$|LPIPS $\downarrow$|FID $\downarrow$|DISTS $\downarrow$|
> |:-|:-:|:-:|:-:|:-:|
> |**Ours**|33.78|**0.005**|**3.04**|**0.0095**|
> |DPS|31.91|0.007|3.59|0.0113|
> |RED-Diff|**37.58**|0.008|4.81|0.0128|
>
> Table 2: $50 \times 50$ box
> |Sampler|PSNR [dB] $\uparrow$|LPIPS $\downarrow$|FID $\downarrow$|DISTS $\downarrow$|
> |:-|:-:|:-:|:-:|:-:|
> |**Ours**|26.45|**0.021**|**9.43**|**0.029**|
> |DPS|25.21|0.023|10.71|0.031|
> |RED-Diff|**28.22**|0.028|13.28|0.037|
>
> - Blind estimation: we also consider a blind experiment using the same settings as in Section 4.3, with the corresponding results Figs. 4 and 5 in the attached link.
>
> From the results, we observe that our model is better than or comparable to other methods, with the added benefit of providing access to the learned density model. Nevertheless, although possible, training is computationally demanding: we used 8 A100 GPUs with a batch size of 64 for 2 weeks to reach 200k iterations. It could be accelerated using sliced score matching [5], a common technique for scaling implicit score matching to higher dimensions, which enables replacing the extra backpropagation step with more memory-efficient (forward-mode) Jacobian vector products.
>
> [1] Negrel, H., Coeurdoux, F., Albergo, M. S., and Vanden-Eijnden, E. Multitask learning with stochastic interpolants. NeurIPS, 2025.
>
> [2] Daras, G., Delbracio, M., Talebi, H., Dimakis, A., and Milanfar, P. Soft diffusion: Score matching with general
> corruptions. TMLR., 2023.
>
> [3] Hoogeboom, E. and Salimans, T. Blurring diffusion models. ICLR, 2023.
>
> [4] Bansal, A., Borgnia, E., Chu, H.-M., Li, J., Kazemi, H.,Huang, F., Goldblum, M., Geiping, J., and Goldstein,
> T. Cold diffusion: Inverting arbitrary image transforms without noise. NeurIPS, 2023.
>
> [5] Song, Y., Garg, S., Shi, J. and Ermon, S. Sliced score matching: A scalable approach to density and score estimation, UAI 2020.

---

> > ### Author Rebuttal · Reviewer_937Z · 2026-04-03
> >
> > Thanks for authors' response. The rebuttal has clarified the motivation for their design choices and provided partial experimental evidence for scalability, but it did not resolve the two core concerns raised in the review. I hope to require substantial additional experiments (covariance simplification ablation, 256×256 high-resolution validation) to justify its claims and practical value.

---

> > > ### Author Response · Authors · 2026-04-03
> > >
> > > We thank the reviewer for engaging with the discussion. We believe the rebuttal has addressed the core concerns more substantially than acknowledged, and we provide further clarification below.
> > >
> > > # Covariance analysis
> > > The experiments we considered are the standard ones for the linear inverse problems family, as used in almost all papers in the diffusion literature [1-4], which are almost all diagonal in some basis. The only one missing is Motion deblurring, which is now included in the PDF (Fig.8) and in the table below (the noise variance is $\sigma_v = 2\times10^{-3}$).
> > >
> > > |Sampler|PSNR [dB] $\uparrow$|LPIPS $\downarrow$|FID $\downarrow$|DISTS $\downarrow$|
> > > |:-|:-:|:-:|:-:|:-:|
> > > |**Ours**|29.42|**0.007**|50.6|**0.083**|
> > > |DPS|28.63|0.02|45.65|0.094|
> > > |RED-Diff|**31.00**|0.02|**44.33**|0.085|
> > >
> > > As the numerical results illustrate, our method is competitive in this OOD covariance. We notice that the FID is higher than the baselines, mainly due to border artifacts. We acknowledge that this is a limitation of our simple frequency-domain processing, and, as we explained in Q5, our frequency-domain degradations are computed using a Fourier transform with periodic boundary conditions, which can introduce border effects.
> > >
> > > We remark that this deblurring type is **not** part of the training set, showing that our model can generalize to other degradations not seen during training (please, see Appendix D for more examples). We will include this new experiment (Motion Deblurring) in the revised manuscript and welcome specific suggestions from the reviewer to enhance our work.
> > >
> > > # Large-scale experiment
> > > We would like to highlight that we have included a new experiment at 192×192 resolution, where we demonstrate that our model can 1) generate high-quality samples for box inpainting, and 2) solve the novel tasks we proposed, such as blind estimation, which validates our density model at scale. This resolution represents a 9× increase in dimensionality over the 64×64 setting (36,864 vs. 12,288 dimensions), which is a substantial scaling step. Importantly, the key qualitative behaviors, namely sample quality, and blind estimation, are all validated at this resolution, and there is no indication that these properties would degrade at 256×256. We acknowledge 256×256 as a natural next step; however, the computational cost of training at that resolution exceeds what is feasible within the rebuttal period, and we spent significantly time scaling the model to a higher-resolution in the rebuttal. We believe the 192×192 results already provide meaningful evidence of scalability beyond the original 64×64 experiments.
> > >
> > > Lastly, we would like to **point out to the reviewer our last response to Reviewer 1, where we contextualize our model in the broad area of energy models**. In particular, this computational challenge is shared by energy-based models more broadly:
> > >    - Thornton et al. [1 in Response to 1] used 8 A100 GPUs at 64×64 resolution (without the dual objective).
> > >    - Balcerak et al. [2 in Response to 1] trained at 64×64 on CelebA with 4 A100s and a batch size of 32; neither [1] nor [2] report total training time.
> > >    - Boffi et al. [3 in Response to 1] required approximately 8 days for the flow matching teacher and 6 days for the shortcut model on ImageNet 64×64.
> > >    - Du et al. [4 in Response to 1] trained at 128×128 using TPU clusters, making direct comparison difficult.
> > >
> > > All these works trained models at lower resolution, supporting our claim that scaling energy-based models to 192×192 is itself a nontrivial contribution.
> > >
> > > # Closing remark
> > > Our paper's contribution lies not in achieving SOTA reconstruction on individual tasks, but in providing a principled framework for learning posterior densities over a family of linear inverse problems with a single model. This unlocks several new applications (energy-guided path, MCMC correction steps, blind estimation, and evaluation of posterior density), which illustrate the practical value of the approach. These applications are unique in the context of inverse problems + diffusion. In addition, thanks to the reviewer's concern on scalability, we could illustrate this on a large-scale experiment as well. We hope the reviewer will consider these strengths, and the additional experiments.
> > >
> > > [1] Chung, H., et al., Diffusion posterior sampling for general noisy inverse problems. ICLR 2023.
> > > [2] Kadkhodaie, Z. and Simoncelli, E. Stochastic solutions for linear inverse problems using the prior implicit in a denoiser. NeurIPS 2021
> > > [3] Mardani, M., Song, J., Kautz, J., and Vahdat, A. A variational perspective on solving inverse problems with diffusion models. ICLR 2024.
> > > [4] Daras, G., Chung, H., Lai, C.-H., Mitsufuji, Y., Ye, J. C.,Milanfar, P., Dimakis, A. G., and Delbracio, M. A survey on diffusion models for inverse problems. arXiv 2024

---

### Official Review · Reviewer_Tgke · 2026-03-11

**Soundness:** 2
**Presentation:** 3
**Significance:** 3
**Originality:** 3
**Overall Recommendation:** 4
**Confidence:** 3

**Summary:**

The paper studies linear inverse problems and proposes learning a normalized energy model via anisotropic denoising. The main idea is to use Anisotropic Covariance Score Matching (A-CSM) so that the model can represent the posterior more explicitly than standard score-based methods. This explicit posterior modeling is used for applications such as adaptive scheduling, MALA-style correction, and blind inverse problem solving. Experiments are conducted on MNIST, CelebA 64x64, and ImageNet64 64x64 for inpainting and Gaussian deblurring, with additional results in the appendix.

**Compliance With Llm Reviewing Policy:**

Affirmed.

**Final Justification:**

The paper studies an interesting direction and introduces a meaningful idea of learning an explicit normalized posterior for linear inverse problems, with capabilities beyond standard score-based methods. The rebuttal was helpful and addressed some of my concerns, especially on tuning fairness, sample selection, and additional experiments. Although some limitations still remain, I keep my original weak-accept score.

**Key Questions For Authors:**

1. The main technical claim of the paper is that the model learns a normalized posterior. Can the authors clarify more precisely why the normalization is valid in practice, and provide stronger empirical or theoretical evidence that the computed posterior values are reliable?

2. It is unclear whether the compared baselines were tuned with effort comparable to the proposed method. Can the authors describe the hyperparameter tuning protocol for each baseline in more detail, including what was tuned, over what range, and whether the same level of effort was used across methods?

3. The current experiments are limited to relatively small-scale settings and the main quantitative results in the main text cover only a limited set of degradations. Do the authors have evidence that the method scales to higher-resolution settings or extends more broadly across linear inverse problems beyond the current covariance/operator family?

4. The paper says that the best sample across steps is selected. Can the authors clarify exactly how this selection is done at test time, and whether any oracle information from the ground truth is used?

**Limitations:**

The limitations section could be strengthened. In particular, the paper should state more clearly that the current evidence is limited to small-scale datasets, so scalability to higher-resolution settings is still unclear. It would also be better to discuss more explicitly that the practical method is restricted to a certain covariance/operator family. It would also help to state more clearly that the normalization claim is not yet fully validated empirically across different covariances.

**Strengths And Weaknesses:**

Strengths:
* The paper studies an interesting direction. Moving from implicit score priors to explicit normalized posteriors is meaningful, and it opens posterior-level uses such as blind inverse solving and MALA-style correction.
* The connection between anisotropic denoising, covariance-conditioned energy modeling, and posterior inference is reasonably clear and technically well motivated.
* The paper includes not only the main benchmark results but also additional analyses such as posterior probability behavior, adaptive scheduling, and dual-vs-single objective training.

Weaknesses:
* It is unclear whether the method scales to higher-resolution settings. Since the current experiments are limited to 28x28 and 64x64, it is hard to know whether both training and posterior sampling remain practical at more realistic resolutions, especially since the paper also notes extra training cost due to double backpropagation.
* The practical formulation seems somewhat restrictive in the operator class it can represent. Since the method uses diagonal covariances in either the spatial or frequency domain, and also rewrites the problem through H^{-1}, it is not clear how broadly it extends to more general linear inverse problems.
* It is unclear whether the compared baselines are tuned with similar effort. Since some baselines required task-specific interventions, more detail on the tuning protocol would make the comparison more convincing. Also, the paper says that the best sample across steps is selected, but it is not clear how this step is chosen.
* The normalization of the energy model is one of the main technical claims of the paper, but this part still needs more clarification. Since downstream applications such as blind solving rely on explicit posterior values, this point is important.
* The paper claims a single model for diverse linear inverse problems, but the main quantitative evidence in the main text is still limited. In addition, the visual quality of the proposed method is not always fully convincing, especially in the super-resolution results.

---

> ### Author Rebuttal · Authors · 2026-03-31
>
> We thank the reviewer for the constructive feedback. Please find our response below, and Figs. in https://anonymous.4open.science/r/rebuttal-211F/rebuttal_images.pdf. Due to space limitations, we point to other reviewers' responses for similar questions.
>
> # W1 Scalability
> Please see response to W2 of Reviewer pAYq
> # W2 Extension to general linear inverse problems
> We thank you for raising this point. First, we would like to clarify that our mathematical framework using $H^{-1}$ is fully general and imposes no condition on the degradation. For cases where the inverse leads to noise of infinite variance (when $H$ is not invertible, e.g., for inpainting), we can approximate those values with a large variance, as is typically done in diffusion models.
>
> Based on this general formulation, we enforce the diagonal constraint, a design choice driven by computational tractability: a general $\Sigma$ would require $\frac{d(d-1)}{2}$ parameters (mentioned in Section 3.3 item $(ii)$), which is prohibitive even for images at $64\times64$. This assumption aligns with recent works using an anisotropic process [Negrel 2025, Daras 2023 in main paper].
>
> Thus, two natural choices arise for $\Sigma$: 1) diagonal in the pixel domain, which covers inpainting and spatially varying noise; and 2) diagonal in the frequency domain, which covers Gaussian deblurring and super-resolution. We emphasize that a broad class of linear inverse problems considered in prior works falls in the union of these two categories. In addition, we can handle _both_ categories with a _single_ network, a unique contribution of our proposed methodology that goes beyond previous works.
>
> Nevertheless, we believe there are alternative ways to extend this approximation by considering low-dimensional parameterizations of $H$, as in [2]. This is the case of several linear inverse problems in practice, where one considers a family of covariances $\Sigma_\phi$ for a low-dimensional parameter $\phi$. This can be accommodated in our framework by letting the network compute $U_\theta(y, \phi) \approx U(y, \Sigma_\phi)$. This requires (straightforward) modifications to the embedding network and the A-CSM objective to handle this low-dimensional parameterization $\phi$.
>
> # W3/Q4 Baselines tuning and best sample selection
> - Hyperparameter tuning: We perform a grid search for each baseline for every task, including NFE, to obtain the best possible performance for each method. The number of NFEs is kept approximately equal across methods to ensure a fair comparison.
> - Sample selection across steps: We clarify that no oracle-based post-processing is used. 'Best sample across steps' refers to early stopping: we tune the number of iterations via ablation, then fix it for all experiments. This particularly applies to RED-Diff, which performs better with fewer noise levels (consistent with Appendix D.2 in [Mardani 2024]). The step count is fixed at test time without using ground-truth information.
> # W4 Reliability and accuracy of the normalization
> Our normalization procedure builds on [Guth 2025 in paper], which has already demonstrated that (isotropic) dual score matching objective allows accurate recovery of normalized energy values in a high-dimensional synthetic setting where the ground truth is available. Our generalization to the anisotropic setting preserves this property. In addition, we verified that our computed energy values reproduce the results in [Guth 2025 in paper] of (approximately) Gumbel-distributed log-probability values and correlation between energy and ''complexity'' of the image (see Figs. 6 and 7 in the attached link).
> # W5 Visual quality and single model claim
> First, we remark that all reported results, including inpainting, Gaussian deblurring, and super-resolution, use a single model trained jointly across all degradation types, which we believe already constitutes meaningful evidence for the claim. The additional AFHQ-Cats experiments at 192×192 further demonstrate generalization to higher resolutions.
>
> We acknowledge that super-resolution results are on par with baselines rather than strictly superior. We attribute this partly to the known limitations of the frequency-domain conditioning, which can introduce border artifacts (see response to Q7 to Reviewer 4). We note, however, that the primary goal of this paper is not to introduce a highly optimized architecture, but rather to establish the formulation and demonstrate the range of capabilities that an explicit normalized posterior enables, such as adaptive scheduling, MALA-style correction, and blind estimation, none of which are available to standard score-based methods. Improving the reconstruction quality, particularly for super-resolution, is a natural direction for future work.
>
> [1] Song, Y., et al. Sliced score matching: A scalable approach to density and score estimation, UAI 2020.
> [2] Chung, H., et al. Parallel diffusion models of operator and image for blind inverse problems. CVPR, 2023

---

> > ### Author Rebuttal · Reviewer_Tgke · 2026-04-01
> >
> > The rebuttal partially addresses my concerns. It is helpful that the authors clarified that the reported inpainting/deblurring/SR results come from a single jointly trained model, that the baselines were tuned per task, and that "best sample across steps" means fixed early stopping without oracle selection; the linked additional results on 192x192 AFHQ and the extra normalization figures are also helpful. However, I still think the normalization claim needs stronger validation, and the practical scope still seems limited to the current covariance/operator family. Also, while the new evidence improves the scalability story, it is still not fully clear to me how broadly the method extends, and the visual quality is still not fully convincing, especially for super-resolution.

---

> > > ### Author Response · Authors · 2026-04-03
> > >
> > > We thank the reviewer for engaging with the discussion and are pleased that we were able to address your concerns partially. We will try to resolve the missing ones.
> > >
> > > # Practical scope
> > > The experiments we considered here are the standard ones for the linear inverse problems family, as used in almost all papers in the diffusion literature [1-4]. In particular, we covered box inpainting (with different box sizes), random inpainting, Gaussian deblurring, and super-resolution. The only one missing is Motion deblurring, which is now included in the PDF (Fig.8) and in the table below (with noise variance $\sigma_v = 2\times10^{-3}$).
> > >
> > > |Sampler|PSNR [dB] $\uparrow$|LPIPS $\downarrow$|FID $\downarrow$|DISTS $\downarrow$|
> > > |:-|:-:|:-:|:-:|:-:|
> > > |**Ours**|29.42|**0.007**|50.6|**0.083**|
> > > |DPS|28.63|0.02|45.65|0.094|
> > > |RED-Diff|**31.00**|0.02|**44.33**|0.085|
> > >
> > > As the numerical results illustrate, our method is competitive in this OOD covariance. We notice that the FID is higher than the baselines, mainly due to artifacts at the border. We acknowledge that this is a limitation of our simple frequency-domain processing, and, as we explained in Q5, our frequency-domain degradations are computed using a Fourier transform with periodic boundary conditions, which can introduce border effects.
> > >
> > > We remark that this deblurring type is **not** part of the training set, showing that our model can generalize to other degradations not seen during training. On the other hand, to consider cases with even stronger blurring degradation, we would need to train a new model with the motion kernel as a possible covariance, which we cannot complete before the rebuttal. We will include this new experiment (Motion Deblurring) in the revised manuscript and welcome specific suggestions for other degradations from the reviewer to enhance our work.
> > >
> > > # Normalization constant
> > > We did not include a dedicated experiment for the normalization constant because the numerical validation of this quantity was already established in the DSM paper [5], which validated the approach by leveraging the isotropic Gaussian distribution at a very high noise level using eq. (10) in the main paper. But to make stronger the argument and further validate that the anisotropic path yields the same result, we extended their experiment to the anisotropic setting. In particular, the target distribution is identical $\frac{1}{2}\mathcal{N}(0, \sigma_1^2 I_d) + \frac{1}{2}\mathcal{N}(0, \sigma_2^2 I_d)$ with $\sigma_1=1$, $\sigma_2=4$, $d=1{,}000$. Instead of a single noise level $t$, each training step draws a random partition of dimensions into two equal groups $(A, B)$ with independent noise levels $t_A, t_B \sim \text{UniformLog}(t_{\min}, t_{\max})$. We include the results in Fig. 9 and the corresponding code (simple_example_aniso.ipynb) at the link provided. We will include this additional experiment in the final paper, and we hope this addresses your concern. We also thanks the reviewer for asking this question, we believe this experiment enhances our paper.
> > >
> > > Furthermore, validation on image datasets is impossible because there is no ground-truth normalization constant for natural images. Hence, the only way to validate the correctness of our model is through proxy experiments, such as the blind estimation experiment: the model trained with dual score matching yields correct estimates of both the noise level and the box size, whereas the model trained with single score matching fails on this task. We believe this provides compelling indirect evidence that the normalization constant is correctly captured.
> > >
> > > Lastly, we would like to point out a bug we found in the Gaussian deblurring results in Table 1, which was reported to Reviewer 4.
> > >
> > > # Closing remarks
> > > We emphasize that the goal of our model is not building a SOTA sampler, but instead learning a posterior density by leveraging the structure from different inverse problems. This unlocks several new applications (energy-guided path, MCMC correction steps, blind estimation, and evaluation of posterior density), which illustrate the practical value of the approach. We now also included an additional OOD motion deblurring task, scalability to 192×192 AFHQ images and a validation of the normalization constant when using an anisotropic scheduling.
> > >
> > > [1] Chung, H., et al., Diffusion posterior sampling for general noisy inverse problems. ICLR 2023.
> > > [2] Kadkhodaie, Z. and Simoncelli, E. Stochastic solutions for linear inverse problems using the prior implicit in a denoiser. NeurIPS 2021
> > > [3] Mardani, M., Song, J., Kautz, J., and Vahdat, A. A variational perspective on solving inverse problems with diffusion models. ICLR 2024.
> > > [4] Daras, G., Chung, H., Lai, C.-H., Mitsufuji, Y., Ye, J. C.,Milanfar, P., Dimakis, A. G., and Delbracio, M. A survey on diffusion models for inverse problems. arXiv 2024
> > > [5] Guth, F., Kadkhodaie, Z., and Simoncelli, E. P. Learning normalized image densities via dual score matching. NeurIPS 2025.

---

### Official Review · Reviewer_pAYq · 2026-03-12

**Soundness:** 3
**Presentation:** 2
**Significance:** 2
**Originality:** 3
**Overall Recommendation:** 4
**Confidence:** 4

**Summary:**

The authors introduce an approach based on energy-based models for learning normalized conditional densities that can be used to solve linear inverse problems.

They extend recent work from unstructured/isotropic noise to anisotropic/colored noise, introducing Anisotropic Covariance Score Matching (A-CSM).

They train and evaluate their model for image inpainting and Gaussian deblurring on the MINST, CelebA (64x64) and Imagenet (64x64) datasets. Their approach is competitive with score-based methods (DPS, DAPS, RED-Diff, and Palette), and also enables some additional capabilities.

**Compliance With Llm Reviewing Policy:**

Affirmed.

**Final Justification:**

The authors provided a solid rebuttal, with a substantial amount of new results. I think this is a quite interesting paper with a pretty neat method. I remain somewhat borderline on it, but leaning slightly towards accept. I have increased my score to "4: Weak accept".

**Key Questions For Authors:**

Questions/suggestions:
- The final paragraph of Section 3.1 is somewhat difficult to follow. Same with the (iii) paragraph of Section 3.3. I also found Section 4.2 overall a bit difficult to follow.

- End of Section 4.2.1, "_Preliminary experiments on a CelebA inpainting task indicate that the energy-guided sampler underperforms predefined schedules in this setting [...] leave exploration of those questions to future research_", could you expand on this?

- Could you apply your method to more high-resolution images than 64x64, for example 256x256? Or, does this get really computationally expensive?

- You train the model on inpainting and Gaussian deblurring, and then evaluate on the same type of tasks. Will the model always have to be trained on the same type of task that it should be applied to?

- Equation 3: Not quite clear what \Sigma_t is here (when you read the paper for the first time and get to this point), could probably give a bit more details and compare it with equation 1?










Minor things:
- Line 41, "linear inverse problems and colored noise, our model, learns" --> "linear inverse problems and colored noise, our model learns"?
- You talk about "Plug-and-play methods" and "Conditional methods" in Section 1, but then in Section 2.2 you describe "Plug-and-play" and "Posterior-based", unify this perhaps?
- Line 165, "in Song et al. (2021a); Guth et al. (2025)" --> "in (Song et al., 2021a; Guth et al., 2025)"? The same also on line 173-174? The citation formatting is a bit inconsistent in a few different places.
- Figure 4, "_the colored border indicates the sampler used to produce the image (green for DPS, red for RED-Diff, orange for ours) and its prior probability (darker shades for higher-probability images), matching the arrows in the left panel_": Brighter shades for higher-probability images? Also, it's quite difficult to distinguish red and orange borders here I think (which images are generated by yours vs RED-Diff?).

**Limitations:**

Yes.

**Strengths And Weaknesses:**

Strengths:
- The proposed method is quite interesting and makes sense overall I think.
- It is competitive with score-based methods, while enabling some additional capabilities thanks to the energy function and normalized density.





Weaknesses:
- The paper could be more well written overall. I found some parts of Section 3 and Section 4 somewhat difficult to follow. Figures could be neater (especially Figure 2).
- The proposed method is not applied to any dataset with more high-resolution images than 64x64, the authors write that "_training energy- based models via (dual) score matching is more computationally intensive than training score-based model_", but it is unclear how big of a problem this is in practice.

---

> ### Author Rebuttal · Authors · 2026-03-31
>
> We thank the reviewer for the positive comments about our paper and the constructive feedback.
> Please find the comments addressed below. Link to anonymous PDF with Figures: https://anonymous.4open.science/r/rebuttal-211F/rebuttal_images.pdf
> # W1 Writing quality.
> We apologize for the lack of clarity and thank the reviewer for flagging these sections. We commit to revising all three sections in the final version. We also include an updated version of Fig. 2 in the attached link (where it is numbered Fig. 1).
> # W2/Q3 Scalability to high resolution
> We thank the reviewer for raising this point. Since our submission, we trained a model on the higher-resolution dataset AFHQ-cats $192\times192$ (following [1]).
> - Sampling: we compare with DPS using 800 NFEs for box inpainting; the results are shown in the following tables, and a subset of examples are in Fig.2 and 3 in the attached link.
>
> Table 1: $30 \times 30$ box
> |Sampler|PSNR [dB] $\uparrow$|LPIPS $\downarrow$|FID $\downarrow$|DISTS $\downarrow$|
> |:-|:-:|:-:|:-:|:-:|
> |**Ours**|33.78|**0.005**|**3.04**|**0.0095**|
> |DPS|31.91|0.007|3.59|0.0113|
> |RED-Diff|**37.58**|0.008|4.81|0.0128|
>
> Table 2: $50 \times 50$ box
> |Sampler|PSNR [dB] $\uparrow$|LPIPS $\downarrow$|FID $\downarrow$|DISTS $\downarrow$|
> |:-|:-:|:-:|:-:|:-:|
> |**Ours**|26.45|**0.021**|**9.43**|**0.029**|
> |DPS|25.21|0.023|10.71|0.031|
> |RED-Diff|**28.22**|0.028|13.28|0.037|
>
> - Blind estimation: we also consider a blind experiment using the same settings as in Section 4.3, with the corresponding results Figs. 4 and 5 in the attached link.
>
> From the results, we observe that our model is better than or comparable to other methods, with the added benefit of providing access to the learned density model. Nevertheless, although possible, training is computationally demanding: we used 8 A100 GPUs with a batch size of 64 for 2 weeks to reach 200k iterations. It could be accelerated using sliced score matching [5], a common technique for scaling implicit score matching to higher dimensions, which enables replacing the extra backpropagation step with more memory-efficient (forward-mode) Jacobian vector products.
>
> # Q2 Expand on "Preliminary experiments on a CelebA inpainting task indicate that the energy-guided sampler underperforms predefined schedules in this setting [...]"
>
> Thanks for raising this point. In theory, with a perfect score $\nabla_y U_\theta = \nabla_y U$ and an infinite number of sampling steps, all schedules $(\Sigma_t)_t$ are equivalent and lead to a sample from $p(x)$. In practice, different schedules yield different approximations to the score and discretization errors. The energy-guided schedule is a heuristically motivated schedule that may or may not outperform predefined isotropic schedules in this regard. However, our current anisotropic energy model is restricted to diagonal covariances in the spatial or frequency domain, limiting the energy-guided schedule to a subspace of all possible covariances. Lifting this limitation could enable taking full advantage of energy guidance to define better covariance schedules.
>
>
> # Q4 Inference on out-of-distribution degradations.
> Not necessarily. As demonstrated in Appendix~E.2.2, our model
> supports out-of-distribution sampling, which means it can generalize beyond the specific degradations
> seen during training. However, this is an empirical question, and there are no theoretical guarantees that the energy model generalizes to out-of-distribution covariances, as is the case with any learning-based approach.
>
> # Q5 Clarifications of what is $\Sigma_t$ in eq. 3:
> We apologize for the lack of clarity. In Eq.(3), $\Sigma_t$ denotes the generalization of the isotropic noise $\sigma_t$; it does not have any particular structure, but instead represents just a different forward process.
> In fact, this is the noising process used in recent work, such as [1-4].
>
> # Minor comments.
> Thank you for catching these typos and areas for improvements. We will correct them in the final version and follow your suggestions for Fig. 4 in the main paper.
>
> [1] Negrel, H., Coeurdoux, F., Albergo, M. S., and Vanden-Eijnden, E. Multitask learning with stochastic interpolants. NeurIPS, 2025.
>
> [2] Daras, G., Delbracio, M., Talebi, H., Dimakis, A., and Milanfar, P. Soft diffusion: Score matching with general
> corruptions. TMLR., 2023.
>
> [3] Hoogeboom, E. and Salimans, T. Blurring diffusion models. ICLR, 2023.
>
> [4] Bansal, A., Borgnia, E., Chu, H.-M., Li, J., Kazemi, H.,Huang, F., Goldblum, M., Geiping, J., and Goldstein,
> T. Cold diffusion: Inverting arbitrary image transforms without noise. NeurIPS, 2023.
>
> [5] Song, Y., Garg, S., Shi, J. and Ermon, S. Sliced score matching: A scalable approach to density and score estimation, UAI 2020.

---

> > ### Author Rebuttal · Reviewer_pAYq · 2026-04-04
> >
> > Thank you for the response.
> >
> > I have read the other reviews and all rebuttals.
> >
> > The other reviewers raise some valid concerns, we all seem to have a quite similar view of the paper I think.
> > The authors provided a solid rebuttal, with a substantial amount of new results.
> >
> > I think this is a quite interesting paper with a pretty neat method, but I remain quite borderline on it.
> >
> > _"Nevertheless, although possible, training is computationally demanding: we used 8 A100 GPUs with a batch size of 64 for 2 weeks to reach 200k iterations"_: This sounds like a lot. Is your method significantly more computationally intensive to train than standard score-based models?

---

> > > ### Author Response · Authors · 2026-04-04
> > >
> > > We thank the reviewer for reading all reviews and rebuttals, and for recognizing the substantial new results provided. We are glad the reviewer finds the method interesting.
> > >
> > > Yes, our method is more computationally intensive to train than standard score-based models. This overhead primarily comes from double backpropagation, which is inherent to energy-based formulations and is widely acknowledged in prior work [1–4]. In this sense, the increased cost is not specific to our method, but rather a known trade-off when learning energy models. That said, we further contextualize below that our model has a computational cost comparable to prior work when accounting for model size and resolution.
> > >
> > > 1. We use the Song architecture from EDM ($\approx 70$M parameters), which is substantially larger than the network used in the original DSM paper [5] ($\approx 13$M parameters). The larger architecture naturally increases training time, while also yielding improved performance. For reference, EDM training at 64×64 takes approximately 4 days on 8 V100 GPUs ($\approx 4$ A100s) with a batch size of 256 (see https://github.com/NVlabs/edm). Scaling to 192×192 increases the pixel count by 9×, which significantly increases the computational cost, either in terms of GPU count or total training time.
> > >
> > > 2. This computational challenge is shared by energy-based models more broadly:
> > >    - Thornton et al. [1] used 8 A100 GPUs at 64×64 resolution (without the dual objective).
> > >    - Balcerak et al. [2] trained at 64×64 on CelebA with 4 A100s and a batch size of 32; neither [1] nor [2] report total training time.
> > >    - Boffi et al. [3] required approximately 8 days for the flow matching teacher and 6 days for the shortcut model on ImageNet 64×64.
> > >    - Du et al. [4] trained at 128×128 using TPU clusters, making direct comparison difficult.
> > >
> > > 3. Importantly, the inference time of our method is comparable to DPS and standard diffusion samplers, so the computational overhead is concentrated entirely in training and does not affect sampling time. For completeness, we report the runtime and computational cost for sampling a batch of 16 images at 64×64 on Gaussian deblurring:
> > >
> > > | Sampler  | Time [sec] | Computational cost (NFE) |
> > > |----------|------------|--------------------------|
> > > | Ours     | 139        | $(N_p)$ NFEs + $(N_p)$ gradients w.r.t. input (energy) |
> > > | DPS      | 167        | $N$ NFEs + $N$ gradients w.r.t. denoiser |
> > > | RED-Diff | 22         | $N$ NFEs |
> > >
> > > We acknowledge this as a limitation and agree that these details could have been clarified more explicitly in the main paper; we will include this discussion in the final version, and we thank the reviewer for raising this point.
> > >
> > > All in all, we believe the training cost is justified by the unique capabilities our framework provides. Specifically, learning a full posterior density enables: (i) blind estimation of degradation parameters, (ii) computation of normalization constants and posterior probabilities, (iii) energy-guided/adaptive sampling paths, and (iv) MCMC correction steps for improved posterior sampling. None of these are possible with standard score-based approaches such as DPS or RED-Diff.
> > >
> > > Furthermore, our work demonstrates that it is now possible to learn posterior densities at this scale, which we believe opens a new direction beyond the specific inverse problems considered here. We hope the reviewer will weigh these contributions against the computational overhead.
> > >
> > > [1] Thornton, J., Béthune, L., Zhang, R., Bradley, A., Nakkiran, P., & Zhai, S. *Composition and Control with Distilled Energy Diffusion Models and Sequential Monte Carlo.* AISTATS 2025
> > >
> > > [2] Balcerak, M., Amiranashvili, T., Terpin, A., Shit, S., Bogensperger, L., Kaltenbach, S., Koumoutsakos, P., and Menze, B. *Energy Matching: Unifying Flow Matching and Energy-Based Models for Generative Modeling.* NeurIPS 2025
> > >
> > > [3] Ai, X., He, Y., Gu, A., Salakhutdinov, R., Kolter, J. Z., Boffi, N. M., & Simchowitz, M. *Joint Distillation for Fast Likelihood Evaluation and Sampling in Flow-based Models.* ICLR 2026
> > >
> > > [4] Du, Y., Durkan, C., Strudel, R., Tenenbaum, J. B., Dieleman, S., Fergus, R., & Grathwohl, W. S. *Reduce, Reuse, Recycle: Compositional Generation with Energy-Based Diffusion Models and MCMC.* ICML 2023
> > >
> > > [5] Guth, F., Kadkhodaie, Z., and Simoncelli, E. P. *Learning normalized image densities via dual score matching.* NeurIPS 2025

---

### Decision · Program_Chairs · 2026-04-30

**Decision:**

Accept (regular)

**Comment:**

This paper proposes a new energy-based model trained using denoising score matching with a covariance-based regularization for linear inverse problems. The paper originally received 2xWeakReject and 2xWeakAccept. The main concerns include scalability to high resolution, extension to general linear inverse problems, validation of the oversimplified assumption, the way of choosing the learned covariance, etc. The authors have provided rebuttals. Three reviewers mention that most of their concerns have been well addressed and two reviewers improve the rating. One reviewer still questions about the computational cost and the choice of learned covariance. The authors have responded in detail in the discussion phase. Considering the rebuttal and discussions from all reviewers, ACs recommend accepting this paper. The authors are suggested to carefully revise the paper and incorporate newly conducted experiments according to the comments and discussions.